# SORFPP: Enhancing rich sequence-driven information to identify SEPs based on fused framework on validation datasets

**Hongqi Feng[1], Qi Nie[1], Sen Yang[1,2]***

**1** School of Computer Science and Artificial Intelligence Aliyun School of Big Data School of Software, Changzhou University, Changzhou, China, **2** The Affiliated Changzhou No.2 People's Hospital of Nanjing Medical University, Changzhou, China

* ys@cczu.edu.cn

## Abstract

### Background

Genome sequencing has enabled us to find functional peptides encoded by short open read frames (sORFs) in long non-coding RNAs (lncRNAs). sORFs-encoded peptides (SEPs) regulate gene expression, signaling, and so on and have significant roles, unlike common peptides. Various computational methods have been proposed. However, there is a lack of contributive features and effective models. Therefore, a high-throughput computational method to predict SEPs is needed.

### Results

We propose a computational method, SORFPP, to predict SEPs by mining feature information from multiple perspectives in an experimentally validated dataset from TranLnc. SORFPP fully extracts SEP sequence information using the protein language model ESM-2 and curated traditional encoding, including QSOrder, k-mer, etc. SORFPP uses CatBoost to solve the sparsity problem of traditional encoding. SORFPP also analyzes ESM-2 pre-training characterization information with the Self-attention model. Finally, an ensemble learning framework combines the two models and their results are fed into Logistic Regression model for accurate and robust predictions. For comparison, SORFPP outperforms other state-of-the-art models in Matthew correlation coefficient by 12.2%-24.2% on three benchmark datasets.

### Conclusion

Integrating the ensemble learning strategy with contributive traditional features and the protein language encoding methods shows better performance. Datasets and codes are accessible at https://doi.org/10.6084/m9.figshare.28079897 and http://111.229.198.94:5000/.

**Data availability statement:** Publicly available datasets were analyzed in this study. Codes and data are available at https://github.com/OR2513/SORFPP and http://111.229.198.94:5000/. Data is also available on Figshare. DOI: 10.6084/m9.figshare.28079897 (https://figshare.com/articles/dataset/_/28079897). As these datasets are publicly available, there is no need to request access from the authors.

**Funding:** This research was funded by Natural Science Foundation of Jiangsu Province of China (Grant No. BK20230626, funders: S.Y), partly supported by the open funds of the State Key Laboratory of Plant Environmental Resilience (Grant No. SKLPERKF2401, funders: S.Y), Supported by the Open project of State Key Laboratory of Animal Biotech Breeding (Grant No. 2024SKLAB6-1, funders: S.Y), Postgraduate Research & Practice Innovation Program of Jiangsu Province (Grant No. KYCX23_3069, funders: Q.N) and Fourth Batch of Leading Innovative Talents Introduction and Training Projects under the Longcheng Talent Plan in Changzhou City (Basic Research and Innovation) (Grant No. CQ20230086, funders: S.Y), and also supported by Changzhou Sci&Tech Program (Grant No. CJ20241083, funders: S.Y).

**Competing interests:** The authors have declared that no competing interests exist.

## 1. Introduction

High-throughput RNA sequencing (RNA-seq) has facilitated the large-scale and diverse identification of non-coding RNA families, including long non-coding RNA (lncRNA), microRNA (miRNA), Piwi-interacting RNA (piRNA) and circular RNA (circRNA)[1,2]. Long non-coding RNAs (lncRNAs) are a class of heterogeneous RNAs longer than 200 nucleotides (nt) and lacking coding potential. Since the discovery of lncRNAs in the 1990s, lncRNAs have been considered junk transcription [3,4]. In 2007, with the help of high-throughput sequencing technologies, the ENCODE project discovered many non-coding elements with biochemical functions, which significantly overlap with lncRNA [5]. In recent years, with the increase in lncRNA-related research and a deeper understanding of physiological and pathological processes, studies have shown that lncRNAs have coding potential [6,7]. An open reading frame (ORF) generally refers to the nucleotide region between the start codon and the stop codon, and ORFs shorter than 300 codons are referred to as short open reading frames (sORFs) [8]. Increasing evidence suggests that lncRNAs contain sORFs that can be translated into peptides [9,10]. Functional peptides encoded by short open reading frames (sORFs-encoded small peptides, SEPs) within lncRNAs can play a role in various functions [11], including cell signaling, morphogenic regulators, partner proteins, and so on [12]. Due to the small molecular weight of SEPs, they often are overlooked. Exploring these peptides is crucial.

Currently, many lncRNAs have been identified based on transcriptome sequence, and some databases have been developed to describe their genomic annotations [13], tissue-specific expression [14,15], cellular regulation [16], and so on. However, there is a shortage of a thorough and confirmed dataset collection on peptides encoded by lncRNA [17]. TransLnc employed six biological methods to identify SEP sequences, including m6A, IRES, whole-genome translation analysis, ribosome profiling, and mass spectrometry analysis, thereby constructing a new comprehensive resource database. In this paper, research is conducted using a new comprehensive resource database, TransLnc [17]. TransLnc currently records approximately 583,840 peptides encoded by 33,094 lncRNAs. It integrates six types of direct and indirect evidence that can prove the coding potential of lncRNAs, with 65.28% of peptides having at least one kind of evidence.

Due to the short length and small size of sORFs and SEPs, biological experiments face numerous limitations, such as time-consuming processes, low efficiency, high costs [18], etc. With the rapid development of machine learning algorithms, machine learning has played crucial roles in various aspects, including lncRNA-disease associations [19], identification of cell-penetrating peptides [20], etc. Furthermore, it can provide a powerful reference for biological experiment validation, saving a significant amount of time and cost to accelerate research. Currently, the recognition of encoded peptides based on machine learning is still in the early stages. CRITICA [21], CPC2 [22] and PhyloCSF [23] employ alignment methods to distinguish between mRNA and lncRNA, which can be used for identifying encoded peptides. However, these alignment-based methods heavily rely on pre-existing data, limiting their effectiveness. The predicted results will be directly impacted if there is a large gap between new and alignment data. Another method is non-alignment-based methods, which solely rely on the sequence's intrinsic information. In comparison to alignment-based methods, this method is more flexible and versatile. MiPepid is a tool specifically designed to identify peptides, created by Zhu *et al.* [24] with logistic regression (LR) model by 4-mer features. In comparison to tools like CPC [25], CPC2 [22] and CPAT [26], MiPepid not only exhibits better performance in predicting traditional-sized proteins but also demonstrates excellent performance in the identification of peptides. Tong *et al.* [27] developed a feature engineering tool called CPPred, utilizing eight RNA-sequence-based and protein-sequence-based features

collected from CPAT and CPC2. CPPred incorporated CTD features and employed SVM for the identification of coding RNA. Furthermore, CPPred effectively discriminates between coding RNA and non-coding RNA with lengths less than 303 nt. Besides, Zhang *et al.* [28] proposed DeepCPP model, which utilized the CPPred dataset to extract and combine different sequence compositional features, and then newly introduced nucleotide bias descriptors with the mDS feature selection method to filter the contributive features into a CNN model. Additionally, CPE-SLDI [29] is the first method to address the class imbalance issue by employing an oversampling strategy. This strategy helps improve the performance in identifying sORFs.

However, these methods have not explicitly focused on identifying sORFs within lncRNAs. There are differences in the length and codon composition of sORFs belonging to different RNAs [30]. Additionally, due to the shorter growth cycles of plants, extracting sORFs encoded peptides from plant gene sequences is less challenging than from animal gene sequences. Therefore, Zhao *et al.* [31] proposed a new method for sORF identification in plants named sORFplnc, which utilized features such as Hexamer scores and a resampling strategy on imbalanced plant datasets. Chen *et al.* [18] incorporated sORF-encoded peptides from plants into the dataset for identification and employed a novel MCSEN model for feature extraction and utilized PCA dimensionality reduction to develop a new method for sORFs and SEPs identification, named sORFpred. However, machine learning relies on data to make decisions, and it is crucial to consider the variations in training data when applying these techniques. While there are differences between animal and plant data, it is critical to acknowledge that solely focusing on plant sORFs may offer limited animal insights. Therefore, a machine-learning model must be constructed to identify peptides translated from animal lncRNA.

Many computational methods have been proposed, but they still face limitations in predicting SEPs. 1) Datasets: In previous studies, nucleotide sequences of sORF are used to construct encoding features, such as CPPred, CPAT, CPC2, sORFplnc and DeepCPP. But these encoding methods lack peptide information, which can result in less accurate predictions of sORF. The sORFpred method constructed datasets from plant data for nucleotide and amino acid sequences. However, there are differences between plant and animal data, which make the prediction of sORFpred in animal sORFs and SEPs only as a reference. In brief, the datasets utilized by these methods are singular and cannot fundamentally describe the characteristics of SEPs. Moreover, the authenticity of these datasets needs further verification. 2) Features: for feature extraction, the alignment-based feature relies on the inherent data and lacks a deep exploration of the feature information in the sequence data. In non-alignment-based methods, most methods utilize traditional feature encoding methods to extract features from either amino acids or nucleotides, such as MiPepid, CPE-SLDI, DeepCPP and sORFplnc. For SEPs sequences, which differ from traditional peptide sequences, the information in these short peptide sequences requires a more in-depth exploration. Although CPPred and sORFpred simultaneously encode features from amino acid and nucleotide perspectives, these methods overlook the structural representation information of proteins. In brief, most existing computational methods predominantly utilize traditional encoding methods to extract features from either amino acid or nucleotide sequences, which results in inadequate research and analysis for recognizing sORFs and SEPs. 3) Models: In previous model constructions, single models are used to process feature encodings, such as MiPepid, CPPred, DeepCPP and CPE-SLDI, but these methods cannot fully utilize the advantages of the extracted feature information. sORFplnc and sORFpred utilized stacking machine learning models, but the predictive results of these methods lack effective interpretability. Therefore, the current models in these methods do not exhibit satisfactory performance and fail to meet the requirements to predict SEPs.

Given these limitations, in this study, we conducted research on an experimentally validated dataset called TransLnc and proposed a new computational method, SORFPP. The

TransLnc dataset includes sequence data for SEP and sORF from three species, including human, mouse and rat. For feature construction, we conducted feature extraction methods from multiple perspectives to capture the relationship between sORFs and encoded peptides. SORFPP fused traditional features from nucleotide and amino acid sequences and fed them into a machine-learning model. To enhance the features of SEP, SORFPP selected the ESM-2 model for protein sequence feature extraction. ESM-2 extracted protein representation information is input into a deep learning model. The construction process of the SORFPP model can be outlined in the following steps: 1) Obtaining and preprocessing the sORF and SEP datasets from TransLnc; 2) Encoding the nucleotide sequences of sORFs by four traditional encoding methods including 3-mer, 4-mer, Fickett and CTD; 3) Encoding the amino acid sequences of SEPs by six traditional encoding methods including AAC, APAAC, QSOrder, PAAC, 2-mer and 4-mer; 4) Combining the traditional feature encoding of nucleotides with the traditional feature encoding of amino acids to obtain fused features; 5) Using a protein language model ESM-2 model to obtain transformer-enhanced feature embeddings of SEPs; 6) Fusing machine learning and deep learning models using an ensemble learning framework to build the SORFPP model; 7) In the model performance assessment, evaluate the SORFPP performance by using seven metrics, including sensitivity (SN), specificity (SP), Matthews correlation coefficient (MCC), accuracy (ACC), precision, F1 score, and area under the ROC curve (AUC). Fig 1 provides an illustration of the comprehensive workflow for SORFPP.

## 2. Data and methods

### 2.1 Data

In previous studies, the types of experimental data were singular, and these datasets lacked precise experimental validation. Therefore, the datasets used in these methods could affect the accuracy of the prediction results. In this paper, the dataset is sourced from a new comprehensive resource, TransLnc, which extends to both translatable lncRNAs and the immune proteome. TransLnc undergoes experimental validation, providing a more reliable foundation for the predictions made in this research [17].

TransLnc initially retrieves lncRNA sequence data for humans, mice, and rats from the GENCODE (human V32 and mouse VM23) and Ensembl databases. And then, TransLnc translates all lncRNA sequences into amino acid sequences using the three-frame translation method and the seqinr package in the R programming language [11,32,33]. ORF sequences that produce more than 10 amino acids will be retained. The minimum ORF length threshold is determined based on the length of known encoded peptides in lncRNA from previous studies [33,34]. Lastly, the obtained nucleotide and amino acid sequences are validated through six direct and indirect methods.

In this experiment, we initially retrieved all nucleotide and amino acid sequences from TransLnc and categorized them into five groups based on the number of supporting evidence. Table 1 documents the classification results of TransLnc. Table 1 presented the original data of TransLnc. TransLnc identified SEP sequences using biological methods. The "Evidence" column in Table 1 indicated the types of biological methods employed. Sequences supported by 1–4 types of evidence are considered positive samples, and sequences lacking any evidence are treated as negative samples. Additionally, we utilized the open-source program CD-HIT [35] with an 80% threshold to eliminate redundant data from the peptide sequence dataset. As a result of the amino acid and nucleotide sequences in TransLnc maintaining a one-to-one correspondence, we filtered out the corresponding nucleotide sequences based on the deduplicated peptide dataset.

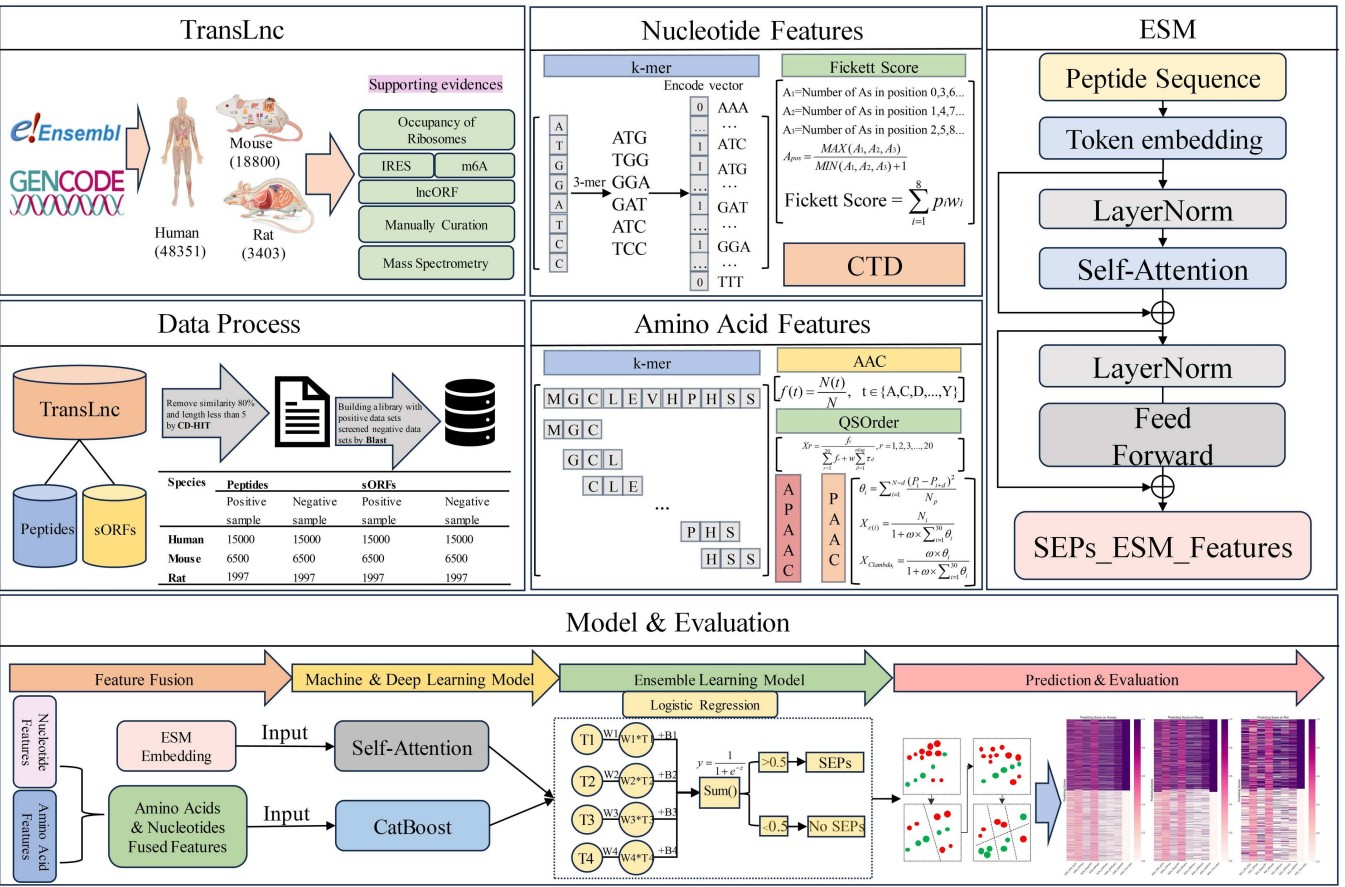

**Fig 1. The SORFPP framework can be concisely outlined in six steps.** 1) Collect, classify and organize data from the TransLnc dataset; 2) Preprocess the organized dataset using CD-HIT and Blast methods to obtain the final experimental dataset; 3) Employ three traditional encoding methods, k-mer, Fickett Score and CTD, for nucleotide sequences; 4) Utilize five traditional encoding methods, k-mer, AAC, QSOrder, PAAC and APAAC, for amino acid sequences; 5) Generate transformer-enhanced feature embeddings for amino acid sequences using the ESM-2 protein language model; 6) Combine and input traditional encoding features of amino acids and nucleotides into the CatBoost model and input ESM-2 features into the Self-Attention model. Finally, the outputs from both methods are merged and inputted into a logistic regression (LR) model for prediction. Then, comparative analysis is conducted with other methods.

**Table 1. Distribution of five supporting evidences in the transLnc dataset.** The "Evidence" column in Table 1 indicated the types of biological methods employed.

| Evidence | Species | | |
|---|---|---|---|
| | Human | Mouse | Rat |
| 4 | 183 | 0 | 0 |
| 3 | 14098 | 7731 | 47 |
| 2 | 73300 | 33414 | 533 |
| 1 | 182458 | 69874 | 2001 |
| 0 | 134861 | 51728 | 13605 |

Moreover, due to the high similarity in the initially screened samples, we employed the Blast [36] method. Initially, positive samples screened using the CD-HIT method were used to create a database, and then the negative samples were compared with the positive sample database. True positive and negative samples were selected using a threshold E-value of 10.

Furthermore, to ensure data balance, we randomly selected data from the positive and negative samples screened using the Blast method at a 1:1 ratio, forming the positive and negative samples used in this paper. Detailed results for the three datasets can be found in Table 2.

## 2.2 Methods

**2.2.1 Feature extraction.** In previous studies, the alignment-based feature relies on the inherent data and lacks a deep exploration of the feature information in the sequence data. In non-alignment-based methods, most methods utilize traditional feature encoding methods to extract features from amino acids or nucleotides, such as MiPepid, CPE-SLDI, DeepCPP and sORFplnc. SEPs sequences differ from traditional peptide sequences.Thus, the information in SEP sequences requires a more in-depth exploration. Although CPPred and sORFpred simultaneously encode features from amino acid and nucleotide perspectives, these methods lack prior knowledge representation of proteins. In brief, most existing computational methods predominantly utilize traditional encoding methods to extract features from amino acid or nucleotide sequences, which results in inadequate feature encoding and feature selection for recognizing sORFs and SEPs.

Therefore, to more accurately distinguish whether sORFs-encoded peptides (SEPs) arebiological activity, this paper employs a multi-perspective feature extraction method to mine SEPs sequence information. Feature fusion is the process of combining data from several contexts into a single entity that can enhance discriminative information and improves the performance of a computational model [37]. On the basis of traditional amino acid feature encoding methods, nucleotide-level sequence features are incorporated. Nucleotide feature extraction methods included 3-mer, 4-mer, Fickett and CTD. Amino acid feature extraction methods included AAC, APAAC, QSOrder, PAAC, 2-mer and 4-mer. Furthermore, to extensively explore the sequence information of SEPs, we utilized the Protein Language Model (PLM) ESM-2 model for feature extraction. This section will detail the advantages of each feature extraction method.

**Nucleotide composition-based feature extraction methods.** This paper evaluated computational tools for identifying lncRNAs proposed by Zheng *et al.*[38] to gain a deeper understanding and investigate nucleotide-related features.

(1) K-mer Features of sORFs Sequence

The csORF-finder model proposed by Zhang *et al.*[39], employed k-mer features to process nucleotide sequences. sORFs sequences consist of Adenines (As), Guanines (Gs), Cytosines (Cs), Thymines (Ts) and unknown bases (N). K-mer features were divided into four categories. Specifically, 1-mer recorded the count of each of the four bases; 2-mer recorded the frequency of occurrences of AA, AG,..., TT; 3-mer recorded the frequency of occurrences of AAA, AAG,..., TTT; 4-mer recorded the frequency of occurrences of AAAA, AAAG,..., TTTT. In this study, SORFPP utilized 3-mer and 4-mer for sequence computation. The 3-mer and 4-mer features were combined into a 750-dimensional vector ($5^3+5^4=750$) [40].

**Table 2. Overview of all the datasets employed in this study.**

| Species | SEPs | | sORFs | |
|---|---|---|---|---|
| | Positive sample | Negative sample | Positive sample | Negative sample |
| Human | 15000 | 15000 | 15000 | 15000 |
| Mouse | 6500 | 6500 | 6500 | 6500 |
| Rat | 1997 | 1997 | 1997 | 1997 |

(2) Fickett TESTCODE Score

CPAT model proposed by Wang *et al.* [26] to distinguish coding RNA and non-coding RNA by the Fickett score feature. The Fickett score was a simple linguistic feature that distinguished protein-coding RNA from non-coding RNA based on a combination of nucleotide composition and bias in codon usage. The Fickett score was obtained by calculating four positional and four composition values (nucleotide content) in the nucleotide sequence. The positional values represented the relative preferences of each base at one codon position compared to another. For example, the positional value of A($A_{pos}$) is calculated as shown in formula (1):

$$A_1 = \text{Number of As in position } 0,3,6...$$
$$A_2 = \text{Number of As in position } 1,4,7...$$
$$A_3 = \text{Number of As in position } 2,5,8... \tag{1}$$
$$A_{pos} = \frac{MAX(A_1, A_2, A_3)}{MIN(A_1, A_2, A_3) + 1}$$

The calculation methods for the positional values of C, G and T were the same as in formula (1). In the calculation process, we determined the composition ratio of each base. These eight composition ratio values were transformed into encoding probabilities (*p*). Each probability was multiplied by the corresponding base's weight (*w*), where the weight values reflected the proportion of time each parameter whether the sequence was encoding or non-coding. SORFPP used Fickett Score to encode a nucleotide sequence to a 1-dimensional vector. The calculation of the Fickett score was shown in formula (2):

$$\text{Fickett Score} = \sum_{i=1}^{8} p_i w_i \tag{2}$$

(3) Composition/Transition/Distribution (CTD)

The Composition Transition Distribution (CTD) features represented global descriptors of transcriptional sequences, including nucleotide composition values, nucleotide transition values and nucleotide distribution values. Originally, the CTD features were used to predict protein folding categories. In 2019, the CPPred model proposed by Tong *et al.*[27] was the first to use CTD features to extract nucleotide sequence information. The CTD features were employed to represent the structural information of sORFs.

The 30 dimensions of CTD features was composed of composition, transition and distribution values. A nucleotide's composition value (C) was the quantity of the selected nucleotide divided by the total number of nucleotides. A nucleotide's transition value (T) represented the percentage frequency of transitions between the four nucleotides in adjacent positions. A nucleotide's distribution value (D) was calculated based on the sequence length, determining its occurrence at five corresponding positions: the first, 25%, 50%, 75% and the last positions.

For example, consider an sORF sequence: ATGGATCCTAGAACCTGTTCTAG-AAG GAGACGC, which contained 10 Adenines (As), 7 Thymines (Ts), 9 Guanines (Gs) and 7 Cytosines (Cs). Then, the corresponding composition values were calculated as follows: 10/33=0.303 for As, 7/33=0.212 for Ts, 9/33=0.272 for Gs and 7/33=0.212 for Cs. When calculating transition values, we consider the adjacent nucleotides (AT, AC, AG, TG, TC and GC) to determine the percentage frequency of transitions. GC represents the frequency of G and C being adjacent or C and G being adjacent. Therefore, the transition values were calculated as follows: AT=4/32=0.125, AC=2/32=0.062, AG=9/32=0.281, TG=2/32=0.062, TC=4/32=0.125, GC=2/32=0.062. Distribution values were computed based on the positions of the first, 25%,

50%, 75% and last positions for each sequence. The five positions corresponding to As were the 1st, 5th, 13th, 24th and 30th. Thus, the distribution values for As are 1/33=0.03, 5/33=0.151, 13/33=0.393, 24/33=0.727 and 30/33=0.909. Similarly, the distribution values for Ts, Gs and Cs are as follows: 0.061, 0.061, 0.273, 0.545, 0.636, 0.09, 0.121, 0.515, 0.788, 0.969, 0.212, 0.212, 0.424, 0.606 and 1. The 20 features were represented by using A0 to A4, T0 to T4, G0 to G4, and C0 to C4[41]. In this study, we used CTD to encode a nucleotide sequence to a 30-dimensional vector.

**Amino acid composition-based feature extraction methods.** This section approaches feature selection from traditional and deep feature encoding methods. Traditional amino acid feature encoding was primarily selected from the iFeature proposed by Chen *et al.* [42]. For deep feature encoding, SORFPP chose the ESM-2 model [43,44] to mine the prior knowledge representational information of sORFs encoded peptides (SEPs) at a deeper level.

(1) Traditional Featuresa. Amino Acid Composition (AAC)

AAC calculated the frequency of each amino acid type in a protein or peptide sequence. The frequency calculation for all 20 natural amino acids (ACDEFGH-IKLMNPQRSTVWY) was shown in Formula (3):

$$f(t) = \frac{N(t)}{N}, \quad t \in \{A, C, D, ..., Y\} \tag{3}$$

Where $N(t)$ was the count of amino acids in class t, and N was the length of the protein or peptide sequence [42]. In this study, we used AAC to encode an amino acid sequence to a 20-dimensional vector.

b. Amphiphilic Pseudo Amino Acid Composition (APAAC)

Unlike traditional AAC, APAAC incorporated the biophysical properties of amino acids, particularly hydrophilicity (PHI) and hydrophobicity (PH). Therefore, the APAAC method was employed to capture more SEP sequence information [45]. The 20 naturally amino acids possessed PHI and PH values that indicated its interaction with water molecules. The PH values spanned from 0 (the least hydrophobic) to 1 (the most hydrophobic), and the PHI values ranged from -1 (the least hydrophilic) to 1 (the most hydrophilic).

The APAAC commences by ascertaining the amino acid composition of the protein sequence and grouping the amino acids into hydrophilic and hydrophobic categories. APAAC calculated the PHI and PH scores for every amino acid in the protein sequence. APAAC was constructed by taking into account the PHI and PH values of adjacent amino acids within a predefined window, centered around a specific position in the SEP sequence. And then, the APAAC feature vector was built by combining each amino acid makeup, PHI and PH scores in the SEP sequence. In all, the amino acid makeup and location-specific attributes of SEP sequences could be obtained by using APAAC encoding method. In this study, we used APAAC to encode an amino acid sequence to a 26-dimensional vector.

c. Quasi-sequence-order (QSOrder)

QSOrder was a feature representation method that describes protein sequences by considering the sequence spacing information between amino acids in the protein sequence. Unlike AAC, QSOrder considered the relationships between adjacent amino acid pairs by defining a series of weights to represent their interactions. Therefore, utilizing QSOrder features better captured the local structural information of protein sequences. This study utilized the encoding characteristics of QSOrder. SEP sequence features were encoded by QSOrder [42], which

transformed the amino acid sequence into a vector symbolising residue. Formula (4) defined a QSOrder descriptor for every amino acid type.

$$Y_{QS} = \frac{N_r}{\sum_{r=1}^{20} N_r + w \sum_{d=1}^{nlag} \tau_d}, r = 1, 2, 3, ..., 20 \tag{4}$$

Here, $N_r$ the standard conversion frequency of amino acids in class r was represented, and a fixed weighting factor $w$ of 0.05 was employed. Formula (5) defined the coupling number of the dth-order sequence.

$$\tau_d = \sum_{i=1}^{L-d} (d_{i,i+d})^2, d = 1, 2, 3, ..., nlag \tag{5}$$

Formula (5), an entry in a specific distance matrix was represented $d_{i,i+d}$. The default value for $nlag$ was 30, and $L$ represented SEP sequences length. In this study, we used QSOrder to encode an amino acid sequence to a 46-dimensional vector.

d. Pseudo-Amino Acid Composition (PAAC)

Chou *et al*. [49] proposed two features called PAAC and APAAC to address the problem of potential sequence information loss. Among them, the PAAC feature encoding method considered the frequency of each amino acid and the influence of sequence order on the amino acid sequence. The calculation method was shown in Formula (6):

$$\begin{cases} \theta_i = \sum_{i=1}^{N-d} \frac{(P_i - P_{i+d})^2}{N_p} \\ X_{c(i)} = \frac{N_i}{1 + \omega \times \sum_{i=1}^{30} \theta_i} \\ X_{Clambda_i} = \frac{\omega \times \theta_i}{1 + \omega \times \sum_{i=1}^{30} \theta_i} \end{cases} \tag{6}$$

Here, $\theta_i$ represented the number of factors related to sequence order, $P_i$ was the property value of the *i-th* amino acid, and $N_p$ was the number of attributes, and $N_i$ was the appearance of the *i-th* amino acid and $\omega$ was a parameter set to 0.05 [46]. In this study, we used PAAC to encode an amino acid sequence to a 23-dimensional vector.

e. K-mer Features of Peptide Sequence

In this paper, k-mer feature encoding was applied to analyze the occurrence frequencies of dipeptides, tripeptides and tetrapeptides in SEP sequences to comprehensively describe the feature information related to the amino acid sequence. In this k-mer method, the amino acid sequence was divided into contiguous subsequences of length k (where k=2,3,4) and then the frequency of each subsequence appearing in the overall sequence was calculated. Through experimental comparisons, 2-mer and 4-mer were ultimately selected for sequence calculations. The calculation method was shown in Formula (7):

$$f(k) = \frac{count(k)}{length(seq) - k - 1} \tag{7}$$

Here, $f(k)$ represented the frequency of occurrence of subsequences of length $k$, and $count(k)$ denoted the frequency of the subsequence appearing in the sequence, and

*length*(*seq*) represented the length of the overall sequence. The term $k-1$ at the end was included to avoid calculating an incomplete subsequence at the sequence's endpoint. In this study, we used k-mer to encode an amino acid sequence to an 841-dimensional vector.

(2) Pre-trained ESM-2 embedding feature

Protein Language Models (PLMs) have emerged as crucial tools for extracting protein-related features. For instance, models like ProtTrans, ESM-1 and ESM-2 have been trained on sequences from the UniRef protein sequence database using masked language modeling objectives. These models were specifically designed for protein feature extraction and enable further exploration of relevant features encoded within peptides.

The Evolutionary Scale Modeling (ESM)[44] protein language model is a method that utilizes deep learning techniques to predict protein structure and function by training an autoregressive neural network on a large-scale protein sequence database and learning the evolutionary patterns of proteins and the relationship between sequence-structure-function. ESM-2 can generate a corresponding hidden vector for a given protein sequence to represent its structural and functional features. ESM is a powerful and versatile protein language model and provides a new perspective and tools for protein science. ESM-2 is pre-trained using the Masked Language Modeling (MLM) task. Its loss function is given by:

$$L_{MLM}(X;\theta) = \mathop{E}_{x \sim X\,mask}\mathop{E}_{i \in mask} logp(xi \mid xj \notin mask;\theta) \tag{8}$$

Here, X represents the training dataset, where a particular symbol denotes amino acid residues at multiple positions on each sample [*Mask*]. In this study, we used ESM-2 to encode an amino acid sequence to a 1280-dimensional vector.

**2.2.2 Methods.** In previous model constructions, single models were used to process feature encodings, such as MiPepid, CPPred, DeepCPP and CPE-SLDI, but these methods could not fully utilize the advantages of the extracted feature information. sORFplnc and sORFpred utilized stacking machine learning models, but the predictive results of these methods lack effective interpretability, such as model evaluation. Therefore, the current models in these methods do not exhibit satisfactory performance. There is room for improvement through the characteristics of machine learning and deep learning model architectures.

In the introduction of the feature section, traditional encoding methods generated numerous zero-valued features. Due to both the amino acid and nucleotide utilize k-mer and other frequency calculation methods, which results in the formation of sparse matrices and leads to suboptimal performance when fed into deep learning models. Moreover, ESM-2 is a Transformer-based protein language model, which can learn the evolutionary patterns of proteins and the relationships between sequence, structure and function from extensive protein sequence data. The highly informative features generated by ESM-2 offer a richer and in-depth inputting feature. Given this wealth of feature information, deep learning models can effectively capture the complex relationships among these features.

In the model selection section, a multi-perspective fusion model method was adopted to explore the advantages of the machine and deep learning models. This paper independently input both types of features into machine learning and deep learning models to get separate prediction results. Finally, employing an ensemble learning framework, we combined the results generated by the machine learning and deep learning models to build a learning model to achieve more accurate predictions.

1. Categorical Boosting (CatBoost)

In this research, our task was to classify and predict the functional peptides encoded by sORFs within lncRNA. Gradient Boosting is an enhancement algorithm used for both regression

and classification tasks, such as CatBoost with two advantages over other machine learning models: 1) handling categorical features, eliminating the need for additional feature engineering before training the model; 2) prediction offset handling, reducing model overfitting and improving prediction performance.

The CatBoost model was a gradient-boosting decision tree method introduced by Prokhorenkova et al. in 2017[47]. CatBoost is a GBDT framework, which employed symmetric decision trees to process the missing and sparse data and capture non-zero features. CatBoost adapted the greedy TS method by incorporating a prior term and assigning weighting factors, as shown in Formula (9):

$$x_k^i = \frac{\sum_{j=1}^{p-1}[x_{\sigma j,k} = x_{\sigma p,k}]Y\sigma j + \alpha \times p}{\sum_{j=1}^{p-1}[x_{\sigma j,k} = x_{\sigma p,k}] + \alpha} \tag{9}$$

In Formula (9), Prokhorenkova $et\ al.$ define $p$ as a prior, typically set to the mean value of the dataset labels, $\alpha$ usually defined as a parameter greater than 0 to ensure the effectiveness of $x_k^i$ when $\sum_{j=1}^{p-1}[x_{\sigma j,k} = x_{\sigma p,k}]$ is 0.

## Algorithm 1: CatBoost of SORFPP

**Input:** $\{(x_i, y_i)\}_{i=1}^n, I, \alpha, L, s, Mode$
```
Initialize:
```
  $\sigma_i \leftarrow$ random permutation of $[1, n]$ for $i = 0..s$

  $S_r(i) \leftarrow 0$ for $r = 0..s, i = 1..n$

  $S'_{r,j}(i) \leftarrow 0$ for $r = 1..s, i = 1..n, j = 1..[log_2 n]$
```
for  t ← 1 to  I  do:
    // Step 1: Calculate gradients
```
    $grad \leftarrow CalcGradient(L, S, y)$;  `// Gradients for current state`

    $grad' \leftarrow CalcGradient(L, S', y)$;  `// Alternate gradients`
```
    // Step 2: Introduce stratified sampling for stability
```
    $r \leftarrow random(1, s)$; `// Stratified sampling to reduce imbalance`
```
    // Step 3: Build decision tree
```
    $T_t \leftarrow BuildTree\left(Mode, grad_r, grad'_r, \sigma r, \{xi\}_{i=1}^n\right)$ `// Using sampled gradients`
```
    // Step 4: Retrieve leaf indices
```
    $leaf_{r,i} \leftarrow GetLeaf(x_i, T_t, \sigma_r)$ for $r = 0..s,\ i = 1..n$
```
    // Step 5: Aggregate gradient updates within leaves
```
    **foreach**  $leaf\ R_j^t$ in $T_t$ **do:**
    if $|leaf\ R_j^t| <$ threshold: `// Avoid small leaf imbalance`
      $SmoothLeaf(leaf\ R_j^t)$ `// Smooth leaf if too few samples`

    $b_j^t \leftarrow -avg\left(grad_0(i)\ for\ i:\ leaf_{r,i} = j\right)$ `// Compute average gradient for the leaf`
```
    // Step 6: Update model states
```
    $S, S' \leftarrow UM(Mode, leaf, T_t, \{b_j^t\}_j, S, S')$ `// Update states using smoothed values`
```
end for
// Final prediction
```
**Return** $F(x) = \sum_{t=1}^{I}\sum_j \alpha\ b_j^t 1_{\{Get\ Leaf(x, T_t, ApplyMode) = j\}}$

2. Logistic Regression (LR)

In this paper, logistic regression (LR) as the final decision-layer model utilizes the linear combination feature. The deep features extracted through the ESM-2 model encompass deeper

semantic information, and the LR model can find effective decision boundaries in effective feature spaces. Moreover, features extracted by traditional encoding methods exhibit collaborative relationships. The logistic regression model can learn these relationships and effectively utilize the information from traditional features.

Due to the simplicity and efficiency of LR, LR is often a favorable choice as the final output layer for ensemble model. The core of the logistic regression model is the S-shaped logistic function for estimating probabilities, as shown in Formula (10):

$$P(Y=1|X) = \frac{1}{1+e^{-(\beta_0+\beta_1 X_1+\beta_2 X_2+...+\beta_k X_k)}} \tag{10}$$

In Formula (10), $P(Y=1|X)$ represents the probability of the amino acid sequence. $\beta_0, \beta_1, ..., \beta_k$ comprised model parameters, including the intercept term $\beta_0$ and coefficients $\beta_1$ to $\beta_k$ for each feature. $X_1, X_2, ..., X_k$ represented the features of the amino acid sequence.

3. Self-attention Model

In the feature extraction phase, we employed the ESM-2 model for in-depth information mining of amino acid sequences. During the pre-training process of ESM-2, a Self-attention mechanism is employed to extract attention maps. Symmetrization and Average Product Correction (APC) are applied to transform the attention maps into the desired format for regression tasks. Thus, to better capture the interactive information among the sequences of SEPs, SORFPP incorporated a self-attention model into the deep learning architecture constructed in this paper [48]. The self-attention mechanism enables the model to capture long-term dependencies between information in the SEPs and selectively assign higher weights to important SEPs sequence information and assign lower weights to other information. Three attention vectors, $Q$, $K$ and $V$, are represented:

$$Q = U_q Z; K = U_k Z; V = U_v Z \tag{11}$$

Here, the query matrix was denoted $Q$, the key matrix was represented $K$, the value matrix was represented $V$, and $U_q, U_k$ and $U_v$ are parameter matrices. The attention weights corresponding to each element are computed using the softmax function. The representation of the attention matrix is given by Formula (12):

$$T = At(Q,K,V) = \text{softmax}(\frac{QK^T}{\sqrt{dk}})V \tag{12}$$

Here, $T$ represents the attention matrix, $dk$ denotes the scaling factor and softmax denotes a column normalization function.

---

### Algorithm 2: Self-attention of SORFPP

**Input:** ESM-2 feature encoding
**Initialize:** Optimizer=Adam, Learning rate=0.001, Epochs=128, Loss-Function= Binary Cross-Entropy

$Attention(Q,K,V) = \text{softmax}(\frac{QK^T}{\sqrt{dk}})V$

$X_1 = Positional\text{-}Encoding\left(Input\text{-}Embedding\left(input\right)\right)$

$X_2 = LayerNorm\left(X_1 + MultiHead\text{-}Attention\left(X_1\right)\right)$

$X_3 = LayerNorm\left(X_2 + Feed\text{-}Forward\left(X_2\right)\right)$

$Y_1 = Positional\text{-}Encoding\left(Output\text{-}Embedding\left(Outputs\right)\right)$

## Algorithm 2: Self-attention of SORFPP

$Y_2 = LayerNorm\left(Y_1 + Masked - MultiHead - Attention\left(Y_1\right)\right)$

$Y_3 = LayerNorm(Y_2 + MultiHead\text{-}Attention(X_3, X_3, Y_2))$

$Y_4 = LayerNorm\left(Y_3 + Feed\text{-}Forward\left(Y_3\right)\right)$

$Y_5 = Linear\left(Y_4\right)$

$Output = Softmax\left(Y_5\right)$

### 4. Construction of SORFPP

This paper presents a computational framework called SORFPP to solve the prediction of SEP. SORFPP used a validated experimental dataset named TransLnc to conduct experiments, which predicted the activity of SEPs by integrating multi-perspective features and models. SORFPP consists of three components: 1) Traditional fusion features of amino acid and nucleotide were processed by using the CatBoost model; 2) ESM-2 model extracts Prior knowledge representation information from SEP sequences. And then, this information was processed by using the self-attention model; 3) The output of CatBoost and self-attention model were combined and processed by using the linear combination feature of a logistic regression model.

For traditional amino acid encoding methods, the feature methods that calculate sequence composition and base frequency made a positive contribution to predicting SEPs. Due to SEPs were obtained through the translation of sORFs. Therefore, this study obtained a traditional fusion feature that combines amino acid and nucleotide perspectives by integrating the nucleotide-level base frequency values with amino acid feature values. The dimension of this fused feature was 1737 ($V_{1737}$), including AAC ($V_{AAC}$), APAAC ($V_{AP}$), PAAC ($V_P$), QSOrder ($V_{QS}$), CTD ($V_{CTD}$), Fickett score ($V_{Fick}$), k-mer of amino acid ($V_k$) and k-mer of nucleotide ($V_k$). And then, the traditional fused feature ($V_{1737}$) will input the CatBoost model.

$$V_{1737} = V_{AAC} \oplus V_{AP} \oplus V_P \oplus V_{QS} \oplus V_k \oplus V_{CTD} \oplus V_{Fick} \oplus V_k, \tag{13}$$

The CatBoost model used the logarithmic loss function to predict SEPs. Here, $y_{true}$ represented true label, and $y_{pred}$ represented probability of prediction. In this study, the depth of trees was set 7 in the CatBoost model. The $leaf_i$ represented i-th tree values. ML_Output represented the probability that a sample belongs to the positive sample when inputting feature is $V_{1737}$.

$$L(y_{true}, y_{pred}) = -y_{true} \otimes \log(y_{pred}) - (1 - y_{true}) \otimes \log(1 - y_{pred}) \tag{14}$$

$$S = \sum_{i=1}^{7} leaf_i \tag{15}$$

$$ML\_Output = P(class = 1 \mid V_{1737}) = \frac{1}{1 + e^{-S}} \tag{16}$$

This study used ESM-2 to encode an amino acid sequence to a 1280-dimensional vector to mine the prior knowledge representational information of SEPs. And then, the ESM features ($V_{1280}$) will input self-attention model. The $V_{1280}$ was transformed into vectors of Query ($Q$), Key ($K$) and Value ($V$). The $dk$ was set 1280 in the self-attention model. DL_Output was a probability distribution, which represents the importance of each input element.

$$Q = V_{1280} \otimes W^Q$$
$$K = V_{1280} \otimes W^K \tag{17}$$
$$V = V_{1280} \otimes W^V$$

$$DL\_Output = Attention(Q, K, V) = \text{softmax}(\frac{QK^T}{\sqrt{dk}})V \tag{18}$$

Combine ML_Output and DL_Output as input ($P_{MD}$) for the logistic regression model. $Z$ represented the result of the linear combination of $P_{MD}$ and the weight coefficients ($W_i$). Here, $B$ represented the model intercept, and n represented the number of samples. And then, $Z$ was transformed into probability of prediction SEPs by using the sigmoid function. Finally, If $P$ was greater than 0.5, sample was classified as SEPs. If $P$ was less than 0.5, sample was classified as not SEPs.

$$P_{MD} = ML\_Output \oplus DL\_Output \tag{19}$$

$$Z = \sum_{i=1}^{n}(W_i * P_{MDi} + B_i) \tag{20}$$

$$P = P(class = 1 | P_{MD}) = \frac{1}{1 + e^{-Z}} \tag{21}$$

The specific steps are illustrated in Fig 2.

**2.2.3 Evaluation.** Evaluation metrics were numerical indicators of model performance. Predicting whether the SEPs were biologically active was a binary classification problem, so we used binary classification metrics to assess the model's effectiveness. In this paper, we employ several commonly used metrics to evaluate SORFPP, such as Accuracy (ACC), Precision (PRE), F1 Score, Area under the ROC curve (AUC), Sensitivity (SN), Specificity (SP) and Matthew's correlation coefficient (MCC). The false positive and true positive rates were represented FPR and TPR. The ROC curve had the FPR on the horizontal axis and the TPR on the vertical axis. The formulas are as follows:

$$Sensitivity = \frac{TP}{TP + FN} \tag{22}$$

$$Specificity = \frac{TN}{FP + TN} \tag{23}$$

$$MCC = \frac{(TP \times TN) - (FP \times FN)}{\sqrt{(TP + FN)(TN + FP)(TP + FP)(TN + FN)}} \tag{24}$$

$$Accuracy = \frac{TP + TN}{TP + FP + FN + TN} \tag{25}$$

$$Precision = \frac{TP}{TP + FP} \tag{26}$$

$$F1 = \frac{2 \times TP}{2 \times TP + FP + FN} \tag{27}$$

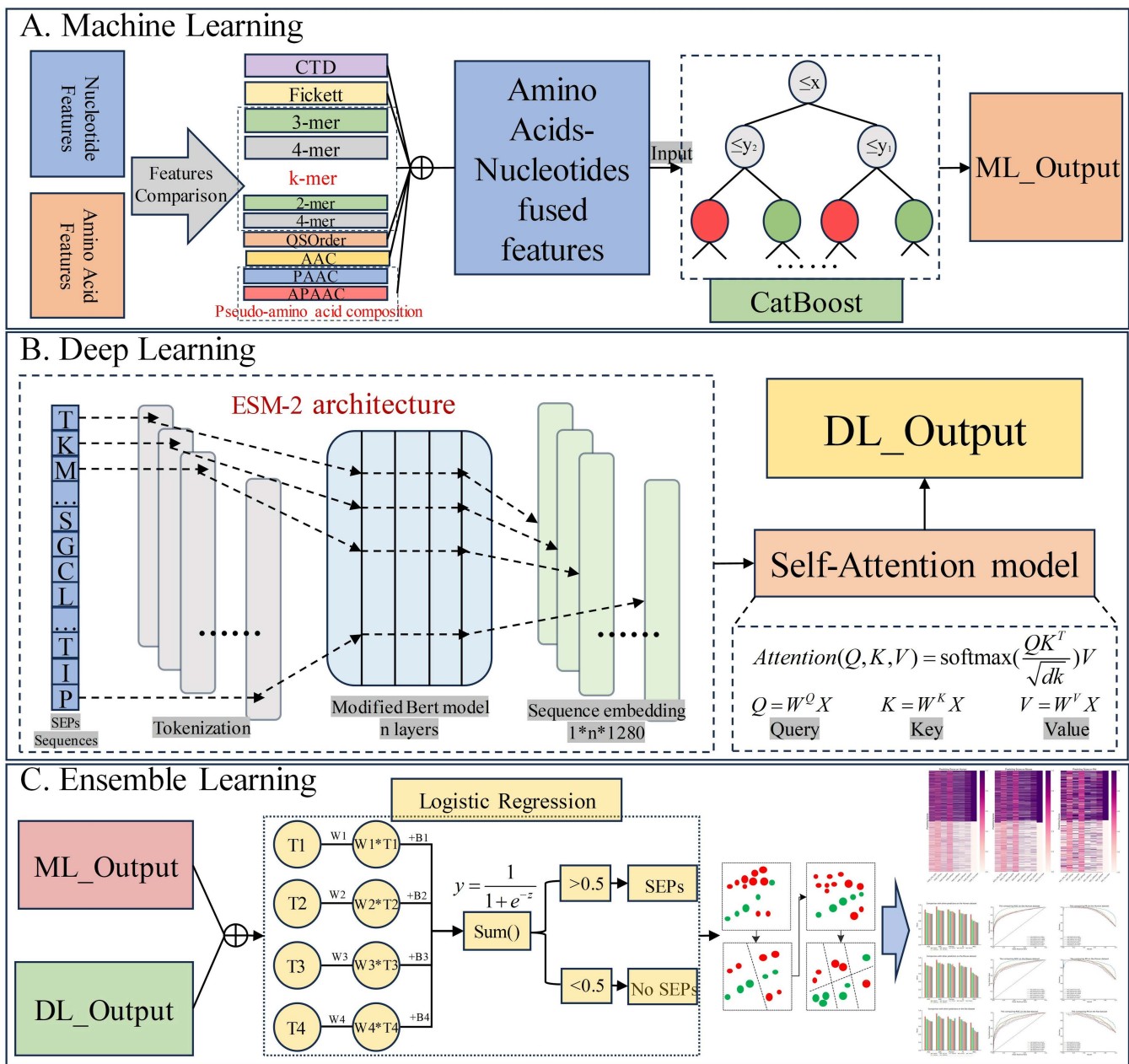

**Fig 2. SORFPP Framework:** A) Machine Learning: Ten encoding methods for nucleotide and amino acid sequences are compared and selected through a multi-perspective method. Subsequently, these ten feature encodings are fused and input into the CatBoost model to obtain machine learning outputs. B) Deep Learning: Coding peptide sequences are input into the protein language model ESM-2. These features are input into the Self-Attention model to obtain deep learning outputs. C) Ensemble Learning: The outputs from machine learning and deep learning are combined and input into the logistic regression model, serving as the final model to produce prediction results.

The TP represented the count of instances where the predicted and actual values were SEPs, TN represented the count where both the predicted and actual values were not SEPs and FP represented the count where the predicted value is SEPs, but the actual value was not SEPs. FN represented the count where the predicted value was not SEPs, but the actual value was SEPs.

## 3. Results

### 3.1 Features comparison from different perspectives

Selecting the appropriate feature extraction methods was crucial and helpful to build predictive model. Insufficient comparison of encoding features may make the advantage of each feature ambiguous. Therefore, this study selected feature methods that were helpful for predicting SEPs by conducting feature extraction from multiple perspectives. This section presented the results of feature comparison and selection.

For the traditional feature extraction of the amino acid sequence, 28 traditional amino acid features were compared, including k-mer (k=2, 3, 4), QSOrder, PAAC, AAC, APAAC, DPC, CKSAAP, EAAC, BLOSUM62, BINARY, ZSCALE, GTPC, DDE, CTDD, CKSAAGP, EGAAC, CTDC, CTriad, GDPC, KSCTriad, CTDT, Geary, Moran, GAAC, NMBroto and SOCNumber [42]. After a comprehensive comparison, six feature encoding methods showed excellent, including 2-mer, 4-mer, QSOrder, PAAC, AAC and APAAC. And then, these six features of amino acid were combined into a fused amino acid feature (Amino acid fused feature, AA Feature).

For the traditional feature extraction of the nucleotide sequence, 17 traditional feature encoding methods were compared, including k-mer (k=2, 3, 4), CTD, Fickett, Hexamer, Word2vec, GC content (GC, GC2, GC3), BINARY, EIIP, Entroy, SNCP, z_curve, SPCP and Cumulative Skew. After the comparison, four encoding methods demonstrated well, including 3-mer, 4-mer, CTD and Fickett. Besides, these four features of nucleotide were merged into a fused nucleotide feature (Nucleotide fused feature, NT Feature). Histograms in Fig 3 illustrated the comparison the AA Feature with amino acid traditional features through three datasets. For NT feature, Fig 3 also showed the comparison with nucleotide traditional features by the same way. Detailed comparative results for the three datasets can be found in Table 3 and Table 4.

For accuracy (ACC), precision (PRE), area under the curve (AUC), Matthew's correlation coefficient (MCC), F1 score, sensitivity (SN) and specificity (SP), the comparison results were listed in Table 3. Specifically, in the human amino acid dataset, AA Feature exhibited better performance in comparison with individual features, with improvement of ACC by 0.4%–1.2%, PRE by 0.2%–1.9%, AUC by 0.5%–1.8%, MCC by 0.3%–1.7%, F1 by 0.2%–1.3%, SN by 0.6%–1.5% and SP by 0.2%–0.9%. Moreover, the top six individual features with the higher performance in the mouse and rat amino acid datasets were the same as in the human amino acid dataset. Therefore, these top six features were merged into fused features. In the mouse amino acid dataset, the results of AA Feature exhibited superiority in comparison with individual features, with enhancement of ACC by 0.7%–1.4%, PRE by 0.2%–2.2%, AUC by 0.2%–1.1%, MCC by 1.2%–2.7%, F1 by 0.5%–1.5%, SN by 0.3%–4.6% and SP by 0.5%–3%. In the rat amino acid dataset, the AA Feature illustrated higher evaluation metrics in comparison with individual features, with advancement of ACC by 0.8%–2.1%, PRE by 0.1%–3.9%, AUC by 0.5%–1.7%, MCC by 0.2%–3.4%, F1 by 0.2%–2.4%, SN by 0.2%–1.2% and SP by 0.1%–4.4%.

The comparison results of the encoding feature methods of nucleotide sequence were presented in Table 4. In the human nucleotide dataset, NT feature showed superiority in comparison with individual features, with advancement of ACC by 0.6%–2.7%, PRE by 0.6%–4.2%, AUC by 0.1%–2.2%, MCC by 1.3%–5.6%, F1 by 0.6%–2.5%, SN by 0.4%–2.2% and SP by 0.5%–4%. Moreover, the top four individual features with the better performance in the mouse and rat nucleotide datasets were the same as in the human nucleotide dataset. Therefore, these four features were combined into fused feature. In the mouse nucleotide dataset, the results of NT Feature illustrated higher evaluation metrics in comparison with individual features, with

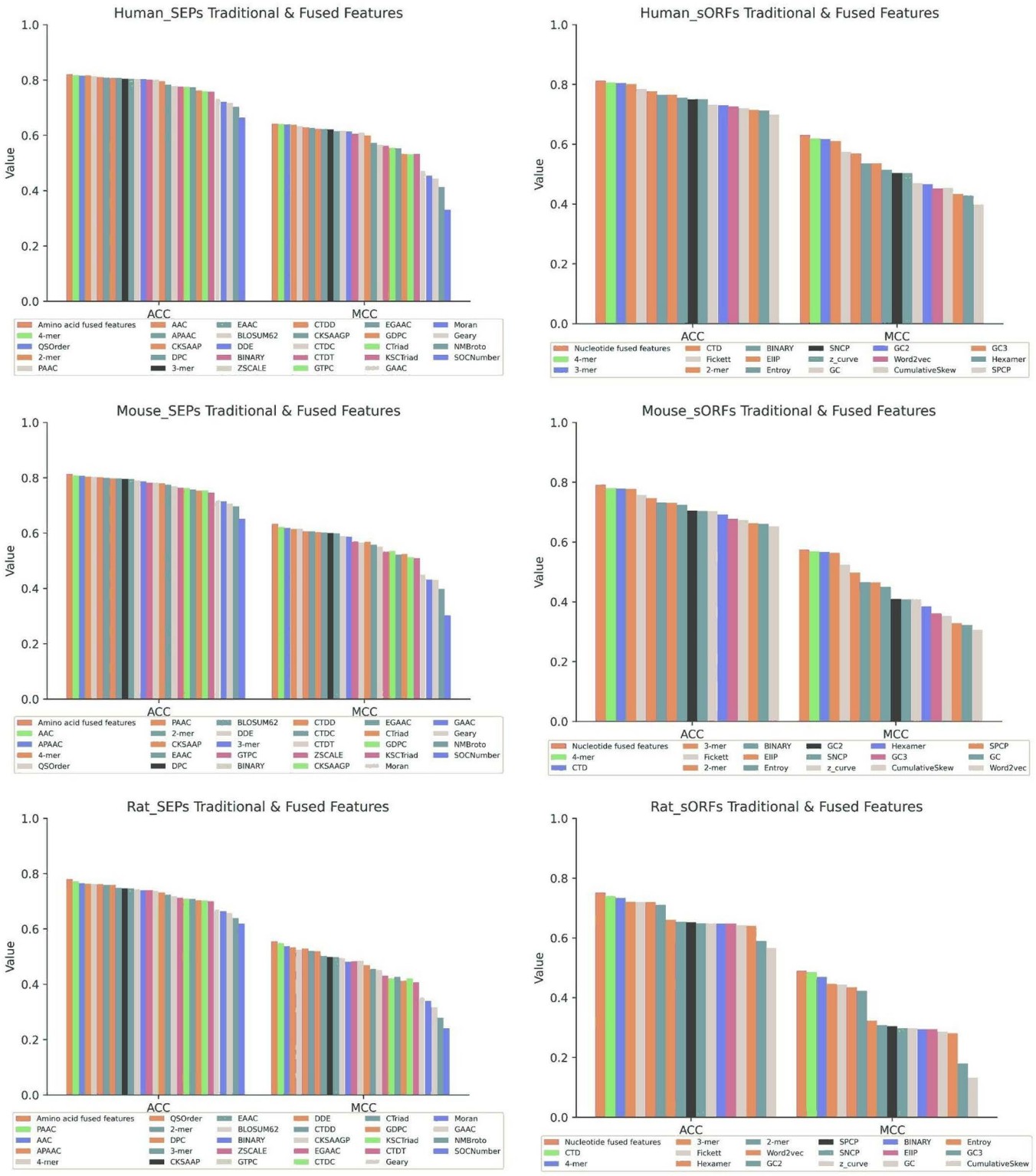

**Fig 3. Comparison results of nucleotide and amino acid features on three datasets.**

**Table 3. Performance of different amino acid feature on three species datasets.**

| Dataset | Feature | ACC | PRE | AUC | MCC | F1 | SN | SP |
|---|---|---|---|---|---|---|---|---|
| SEPs (Human) | **AA Feature** | **0.821** | **0.779** | **0.890** | **0.643** | **0.833** | **0.895** | **0.739** |
| | 4-mer | 0.817 | 0.770 | 0.883 | 0.640 | 0.823 | 0.889 | 0.733 |
| | QSOrder | 0.816 | 0.774 | 0.877 | 0.640 | 0.831 | 0.896 | 0.735 |
| | 2-mer | 0.816 | 0.760 | 0.885 | 0.638 | 0.829 | 0.890 | 0.730 |
| | PAAC | 0.814 | 0.777 | 0.880 | 0.633 | 0.826 | 0.880 | 0.747 |
| | AAC | 0.810 | 0.766 | 0.872 | 0.629 | 0.823 | 0.888 | 0.734 |
| | APAAC | 0.809 | 0.765 | 0.877 | 0.626 | 0.820 | 0.883 | 0.737 |
| | CKSAAP | 0.807 | 0.760 | 0.875 | 0.623 | 0.818 | 0.904 | 0.726 |
| | DPC | 0.807 | 0.760 | 0.878 | 0.623 | 0.819 | 0.889 | 0.726 |
| | 3-mer | 0.805 | 0.770 | 0.880 | 0.622 | 0.820 | 0.880 | 0.732 |
| | EAAC | 0.805 | 0.768 | 0.873 | 0.615 | 0.817 | 0.873 | 0.736 |
| | BLOSUM62 | 0.804 | 0.760 | 0.871 | 0.617 | 0.817 | 0.884 | 0.725 |
| | DDE | 0.803 | 0.759 | 0.870 | 0.614 | 0.815 | 0.879 | 0.730 |
| | BINARY | 0.801 | 0.774 | 0.864 | 0.605 | 0.815 | 0.861 | 0.739 |
| | ZSCALE | 0.800 | 0.757 | 0.864 | 0.609 | 0.815 | 0.883 | 0.718 |
| | CTDD | 0.795 | 0.752 | 0.856 | 0.599 | 0.813 | 0.885 | 0.704 |
| | CKSAAGP | 0.783 | 0.744 | 0.839 | 0.573 | 0.801 | 0.868 | 0.696 |
| | CTDC | 0.779 | 0.737 | 0.848 | 0.567 | 0.794 | 0.862 | 0.697 |
| | CTDT | 0.776 | 0.731 | 0.844 | 0.562 | 0.795 | 0.871 | 0.681 |
| | GTPC | 0.775 | 0.740 | 0.830 | 0.555 | 0.789 | 0.845 | 0.705 |
| | EGAAC | 0.774 | 0.741 | 0.836 | 0.553 | 0.792 | 0.851 | 0.695 |
| | GDPC | 0.762 | 0.723 | 0.823 | 0.532 | 0.777 | 0.839 | 0.688 |
| | CTriad | 0.759 | 0.714 | 0.832 | 0.531 | 0.786 | 0.874 | 0.642 |
| | KSCTriad | 0.758 | 0.707 | 0.831 | 0.533 | 0.786 | 0.884 | 0.632 |
| | GAAC | 0.733 | 0.695 | 0.783 | 0.473 | 0.747 | 0.807 | 0.661 |
| | Moran | 0.721 | 0.680 | 0.781 | 0.454 | 0.750 | 0.835 | 0.606 |
| | Geary | 0.717 | 0.679 | 0.778 | 0.443 | 0.743 | 0.820 | 0.613 |
| | NMBroto | 0.703 | 0.669 | 0.770 | 0.413 | 0.723 | 0.787 | 0.621 |
| | SOCNumber | 0.665 | 0.659 | 0.717 | 0.331 | 0.679 | 0.701 | 0.629 |

*(Continued)*

**Table 3.** (Continued)

| Dataset | Feature | ACC | PRE | AUC | MCC | F1 | SN | SP |
|---|---|---|---|---|---|---|---|---|
| SEPs (Mouse) | **AA Feature** | **0.814** | **0.776** | **0.878** | **0.634** | **0.826** | **0.893** | **0.745** |
| | AAC | 0.807 | 0.764 | 0.867 | 0.622 | 0.821 | 0.887 | 0.726 |
| | APAAC | 0.807 | 0.774 | 0.876 | 0.619 | 0.821 | 0.875 | 0.738 |
| | 4-mer | 0.804 | 0.764 | 0.876 | 0.615 | 0.819 | 0.883 | 0.723 |
| | QSOrder | 0.802 | 0.754 | 0.863 | 0.615 | 0.817 | 0.890 | 0.715 |
| | PAAC | 0.802 | 0.771 | 0.865 | 0.607 | 0.812 | 0.857 | 0.740 |
| | 2-mer | 0.800 | 0.757 | 0.875 | 0.607 | 0.811 | 0.890 | 0.743 |
| | CKSAAP | 0.798 | 0.760 | 0.861 | 0.603 | 0.810 | 0.890 | 0.710 |
| | EAAC | 0.798 | 0.759 | 0.865 | 0.602 | 0.809 | 0.866 | 0.731 |
| | DPC | 0.796 | 0.753 | 0.860 | 0.600 | 0.810 | 0.877 | 0.727 |
| | BLOSUM62 | 0.795 | 0.751 | 0.860 | 0.599 | 0.811 | 0.882 | 0.708 |
| | DDE | 0.790 | 0.748 | 0.850 | 0.589 | 0.807 | 0.876 | 0.705 |
| | 3-mer | 0.787 | 0.746 | 0.851 | 0.587 | 0.800 | 0.850 | 0.712 |
| | GTPC | 0.782 | 0.753 | 0.842 | 0.569 | 0.797 | 0.846 | 0.717 |
| | BINARY | 0.781 | 0.748 | 0.849 | 0.566 | 0.791 | 0.838 | 0.725 |
| | CTDD | 0.780 | 0.740 | 0.838 | 0.568 | 0.801 | 0.874 | 0.683 |
| | CTDC | 0.775 | 0.736 | 0.85 | 0.558 | 0.789 | 0.851 | 0.701 |
| | CTDT | 0.770 | 0.727 | 0.838 | 0.551 | 0.786 | 0.856 | 0.687 |
| | ZSCALE | 0.764 | 0.730 | 0.842 | 0.534 | 0.782 | 0.842 | 0.685 |
| | CKSAAGP | 0.763 | 0.723 | 0.827 | 0.535 | 0.779 | 0.846 | 0.683 |
| | EGAAC | 0.758 | 0.725 | 0.816 | 0.522 | 0.777 | 0.836 | 0.679 |
| | CTriad | 0.754 | 0.701 | 0.822 | 0.525 | 0.782 | 0.884 | 0.623 |
| | GDPC | 0.754 | 0.725 | 0.817 | 0.513 | 0.770 | 0.822 | 0.687 |
| | KSCTriad | 0.746 | 0.698 | 0.814 | 0.510 | 0.778 | 0.880 | 0.610 |
| | Moran | 0.720 | 0.678 | 0.777 | 0.452 | 0.745 | 0.827 | 0.615 |
| | GAAC | 0.715 | 0.695 | 0.756 | 0.432 | 0.731 | 0.772 | 0.658 |
| | Geary | 0.707 | 0.658 | 0.768 | 0.431 | 0.733 | 0.828 | 0.592 |
| | NMBroto | 0.697 | 0.670 | 0.764 | 0.399 | 0.721 | 0.780 | 0.613 |
| | SOCNumber | 0.651 | 0.651 | 0.697 | 0.302 | 0.667 | 0.684 | 0.617 |

*(Continued)*

**Table 3.** (Continued)

| Dataset | Feature | ACC | PRE | AUC | MCC | F1 | SN | SP |
|---|---|---|---|---|---|---|---|---|
| SEPs (Rat) | **AA Feature** | **0.780** | **0.766** | **0.845** | **0.555** | **0.788** | **0.841** | **0.731** |
| | PAAC | 0.772 | 0.747 | 0.840 | 0.548 | 0.786 | 0.829 | 0.715 |
| | AAC | 0.766 | 0.733 | 0.835 | 0.538 | 0.782 | 0.839 | 0.693 |
| | APAAC | 0.763 | 0.728 | 0.831 | 0.533 | 0.777 | 0.834 | 0.694 |
| | 4-mer | 0.762 | 0.765 | 0.840 | 0.524 | 0.764 | 0.830 | 0.730 |
| | QSOrder | 0.762 | 0.727 | 0.828 | 0.529 | 0.778 | 0.837 | 0.687 |
| | 2-mer | 0.759 | 0.730 | 0.831 | 0.521 | 0.770 | 0.830 | 0.692 |
| | DPC | 0.759 | 0.742 | 0.826 | 0.519 | 0.766 | 0.792 | 0.726 |
| | 3-mer | 0.749 | 0.731 | 0.827 | 0.502 | 0.770 | 0.827 | 0.700 |
| | CKSAAP | 0.747 | 0.720 | 0.806 | 0.499 | 0.763 | 0.800 | 0.697 |
| | EAAC | 0.746 | 0.708 | 0.822 | 0.499 | 0.752 | 0.804 | 0.693 |
| | BLOSUM62 | 0.743 | 0.701 | 0.801 | 0.495 | 0.766 | 0.843 | 0.643 |
| | BINARY | 0.740 | 0.724 | 0.804 | 0.481 | 0.753 | 0.785 | 0.694 |
| | ZSCALE | 0.740 | 0.715 | 0.802 | 0.483 | 0.753 | 0.796 | 0.684 |
| | GTPC | 0.738 | 0.703 | 0.795 | 0.484 | 0.756 | 0.818 | 0.660 |
| | DDE | 0.731 | 0.698 | 0.794 | 0.469 | 0.745 | 0.800 | 0.664 |
| | CTDD | 0.724 | 0.694 | 0.784 | 0.455 | 0.744 | 0.804 | 0.645 |
| | CKSAAGP | 0.719 | 0.674 | 0.782 | 0.451 | 0.744 | 0.831 | 0.610 |
| | EGAAC | 0.714 | 0.699 | 0.779 | 0.431 | 0.738 | 0.783 | 0.643 |
| | CTDC | 0.709 | 0.684 | 0.778 | 0.422 | 0.725 | 0.772 | 0.646 |
| | CTriad | 0.709 | 0.672 | 0.764 | 0.427 | 0.738 | 0.819 | 0.598 |
| | GDPC | 0.704 | 0.674 | 0.764 | 0.413 | 0.723 | 0.779 | 0.629 |
| | KSCTriad | 0.703 | 0.662 | 0.766 | 0.420 | 0.733 | 0.822 | 0.586 |
| | CTDT | 0.700 | 0.673 | 0.766 | 0.407 | 0.725 | 0.786 | 0.614 |
| | Geary | 0.672 | 0.640 | 0.716 | 0.354 | 0.703 | 0.780 | 0.566 |
| | Moran | 0.665 | 0.634 | 0.710 | 0.340 | 0.698 | 0.777 | 0.554 |
| | GAAC | 0.658 | 0.651 | 0.703 | 0.317 | 0.674 | 0.699 | 0.616 |
| | NMBroto | 0.639 | 0.633 | 0.694 | 0.279 | 0.660 | 0.690 | 0.588 |
| | SOCNumber | 0.620 | 0.614 | 0.661 | 0.241 | 0.623 | 0.632 | 0.609 |

Note: Amino acid fused feature (AA Feature)

improvement of ACC by 1.2%–4.4%, PRE by 0.4%–3%, AUC by 0.8%–4.5%, MCC by 0.5%–5%, F1 by 0.4%–2.9%, SN by 0.5%–2.8% and SP by 0.1%–2.3%. In the rat nucleotide dataset, the results of NT Feature exhibited superior performance in comparison with individual features, with enhancement of ACC by 1.2%–3.2%, PRE by 0.4%–2.4%, AUC by 0.2%–3.6%, MCC by 0.5%–4.7%, F1 by 0.2%–4.3%, SN by 0.5%–8.3% and SP by 0.3%–2.8%.

SEP sequences had lesser molecular weights in comparison with traditional peptide sequences. Therefore, a multi-perspective feature extraction method was required to enrich SEP sequence information. For traditional feature encoding methods, the AA Feature and NT Feature were combined into the AN Feature. Six traditional amino acid encoding methods were selected, including 2-mer, 4-mer, QSOrder, PAAC, AAC and APAAC. Four traditional nucleotide encoding methods were selected, including 3-mer, 4-mer, CTD and Fickett. And then, the ten feature encoding methods from amino acid and nucleotide were integrated to construct a fused feature that combined the characteristics of amino acids and nucleotides (Amino Acids-Nucleotides fused feature, AN Feature).

**Table 4. Prediction results of different feature encodings on sORFs datasets.**

| Dataset | Feature | ACC | PRE | AUC | MCC | F1 | SN | SP |
|---|---|---|---|---|---|---|---|---|
| sORFs (Human) | **NT Feature** | **0.812** | **0.868** | **0.880** | **0.631** | **0.796** | **0.735** | **0.888** |
| | 4-mer | 0.806 | 0.862 | 0.879 | 0.619 | 0.789 | 0.728 | 0.883 |
| | 3-mer | 0.805 | 0.859 | 0.876 | 0.618 | 0.790 | 0.731 | 0.880 |
| | CTD | 0.801 | 0.855 | 0.874 | 0.610 | 0.785 | 0.725 | 0.877 |
| | Fickett | 0.785 | 0.826 | 0.858 | 0.575 | 0.771 | 0.723 | 0.848 |
| | 2-mer | 0.777 | 0.860 | 0.845 | 0.569 | 0.748 | 0.662 | 0.892 |
| | BINARY | 0.766 | 0.802 | 0.837 | 0.536 | 0.751 | 0.706 | 0.826 |
| | EIIP | 0.766 | 0.802 | 0.837 | 0.536 | 0.751 | 0.706 | 0.826 |
| | Entroy | 0.755 | 0.789 | 0.830 | 0.514 | 0.740 | 0.697 | 0.814 |
| | SNCP | 0.751 | 0.779 | 0.828 | 0.504 | 0.737 | 0.699 | 0.802 |
| | z_curve | 0.751 | 0.779 | 0.828 | 0.504 | 0.737 | 0.699 | 0.802 |
| | GC | 0.732 | 0.774 | 0.801 | 0.470 | 0.710 | 0.656 | 0.808 |
| | GC2 | 0.730 | 0.777 | 0.804 | 0.467 | 0.705 | 0.645 | 0.815 |
| | Word2vec | 0.726 | 0.724 | 0.790 | 0.452 | 0.727 | 0.730 | 0.722 |
| | CS | 0.720 | 0.792 | 0.766 | 0.454 | 0.681 | 0.597 | 0.843 |
| | GC3 | 0.715 | 0.747 | 0.778 | 0.433 | 0.694 | 0.648 | 0.781 |
| | Hexamer | 0.713 | 0.735 | 0.780 | 0.428 | 0.699 | 0.666 | 0.760 |
| | SPCP | 0.699 | 0.715 | 0.768 | 0.399 | 0.687 | 0.660 | 0.737 |
| sORFs (Mouse) | **NT Feature** | **0.792** | **0.843** | **0.855** | **0.574** | **0.764** | **0.699** | **0.868** |
| | 4-mer | 0.780 | 0.839 | 0.847 | 0.569 | 0.760 | 0.694 | 0.867 |
| | CTD | 0.779 | 0.837 | 0.846 | 0.567 | 0.758 | 0.693 | 0.865 |
| | 3-mer | 0.778 | 0.837 | 0.845 | 0.564 | 0.756 | 0.690 | 0.865 |
| | Fickett | 0.758 | 0.813 | 0.810 | 0.524 | 0.735 | 0.671 | 0.845 |
| | 2-mer | 0.747 | 0.778 | 0.823 | 0.498 | 0.732 | 0.692 | 0.803 |
| | BINARY | 0.732 | 0.752 | 0.795 | 0.465 | 0.721 | 0.692 | 0.772 |
| | EIIP | 0.732 | 0.752 | 0.795 | 0.465 | 0.721 | 0.692 | 0.772 |
| | Entroy | 0.724 | 0.743 | 0.791 | 0.450 | 0.713 | 0.686 | 0.762 |
| | GC2 | 0.705 | 0.713 | 0.772 | 0.410 | 0.699 | 0.685 | 0.725 |
| | SNCP | 0.704 | 0.719 | 0.783 | 0.409 | 0.694 | 0.670 | 0.738 |
| | z_curve | 0.704 | 0.719 | 0.783 | 0.409 | 0.694 | 0.670 | 0.738 |
| | Hexamer | 0.692 | 0.690 | 0.754 | 0.385 | 0.694 | 0.698 | 0.687 |
| | GC3 | 0.678 | 0.713 | 0.722 | 0.361 | 0.649 | 0.596 | 0.760 |
| | CS | 0.673 | 0.712 | 0.722 | 0.352 | 0.640 | 0.581 | 0.765 |
| | SPCP | 0.663 | 0.679 | 0.726 | 0.328 | 0.648 | 0.621 | 0.706 |
| | GC | 0.661 | 0.655 | 0.719 | 0.322 | 0.667 | 0.680 | 0.642 |
| | Word2vec | 0.653 | 0.657 | 0.706 | 0.307 | 0.649 | 0.641 | 0.666 |

*(Continued)*

**Table 4.** (Continued)

| Dataset | Feature | ACC | PRE | AUC | MCC | F1 | SN | SP |
|---------|---------|-----|-----|-----|-----|----|----|----|
| sORFs (Rat) | **NT Feature** | **0.752** | **0.776** | **0.810** | **0.490** | **0.723** | **0.693** | **0.808** |
| | CTD | 0.740 | 0.772 | 0.798 | 0.485 | 0.719 | 0.665 | 0.805 |
| | 4-mer | 0.734 | 0.758 | 0.808 | 0.470 | 0.721 | 0.688 | 0.780 |
| | 3-mer | 0.721 | 0.752 | 0.790 | 0.446 | 0.703 | 0.660 | 0.783 |
| | Fickett | 0.720 | 0.767 | 0.774 | 0.443 | 0.680 | 0.610 | 0.805 |
| | Hexamer | 0.720 | 0.705 | 0.788 | 0.434 | 0.730 | 0.758 | 0.683 |
| | 2-mer | 0.711 | 0.719 | 0.769 | 0.423 | 0.706 | 0.693 | 0.730 |
| | Word2vec | 0.661 | 0.650 | 0.720 | 0.323 | 0.673 | 0.700 | 0.623 |
| | GC2 | 0.654 | 0.648 | 0.711 | 0.308 | 0.660 | 0.673 | 0.635 |
| | SPCP | 0.653 | 0.653 | 0.696 | 0.305 | 0.653 | 0.653 | 0.653 |
| | SNCP | 0.649 | 0.646 | 0.692 | 0.298 | 0.652 | 0.658 | 0.640 |
| | z_curve | 0.649 | 0.646 | 0.692 | 0.298 | 0.652 | 0.658 | 0.640 |
| | BINARY | 0.648 | 0.653 | 0.697 | 0.295 | 0.641 | 0.630 | 0.665 |
| | EIIP | 0.648 | 0.653 | 0.697 | 0.295 | 0.641 | 0.630 | 0.665 |
| | GC | 0.643 | 0.634 | 0.685 | 0.286 | 0.654 | 0.675 | 0.610 |
| | Entroy | 0.640 | 0.650 | 0.689 | 0.281 | 0.628 | 0.608 | 0.673 |
| | GC3 | 0.590 | 0.587 | 0.632 | 0.180 | 0.596 | 0.605 | 0.575 |
| | CS | 0.566 | 0.562 | 0.604 | 0.133 | 0.579 | 0.598 | 0.535 |

Note: Nucleotide fused feature (NT Feature); Cumulative Skew (CS)

The protein language model ESM-2 for feature extraction on the peptide sequence was employed. ESM-2 was a pre-trained model trained on a large dataset through unsupervised learning, which can uncover latent feature of a peptide through prior knowledges. Therefore, the extraction method performance of the AN feature, AA feature, NT feature and ESM feature were compared. In the histogram of Fig 4, the red bar represented the AN feature and demonstrated superior performance in evaluation metrics, such as ACC, AUC, F1 and MCC. The blue bars for the ESM feature were slightly lower than the green bar for the AA Feature and higher than the orange bar for the NT Feature. Additionally, ROC curves in Fig 4 were plotted to provide a comprehensive comparison of the four features.

Moreover, scatter plots in Fig 5 were generated by utilizing t-SNE method to better illustrate the performance of AN feature, AA feature and NT feature. Fig 5 illustrated the scatter distributions of the AN Feature in the top row, the AA Feature in the middle row and the NT Feature in the bottom row. In Fig 5, the first-row scatter plots represented the AN Feature and exhibited a more pronounced dispersion. Consequently, the AN Feature had a better performance for prediction SEPs.

In addition, Shapley Additive Explanations (SHAP) was applied to elucidate the importance of the fused features in the model [49]. SHAP provides a way to quantify the contribution of each feature to the model prediction. This holds significant reference value for analyzing the prediction mechanisms of complex models, particularly in our research, where understanding the impact of each feature on the model output is a crucial step [50].

The Left Column of Fig 6 displayed the SHAP value distribution for the top 20 features, where red dots represented positive correlation and blue dots represented negative correlation. The larger the dot size, the more significant the feature's impact on the model. Through this graph, we observe that the top 20 features contain both features from contrastive learning and features from pre-trained models. The Right Column of Fig 6 provided the average

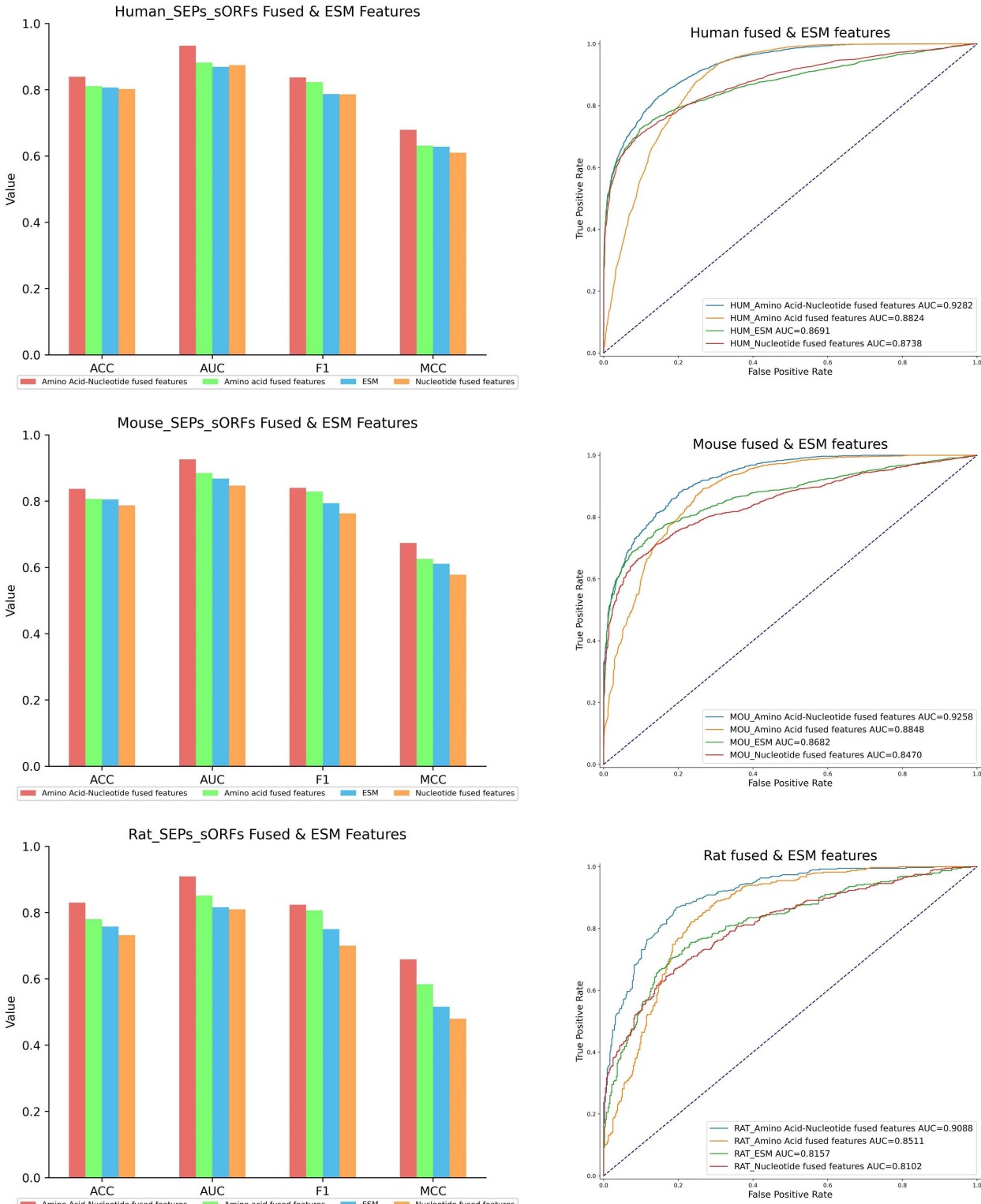

**Fig 4. Comparison of AN feature, AA feature, NT feature and ESM feature on three species datasets.**

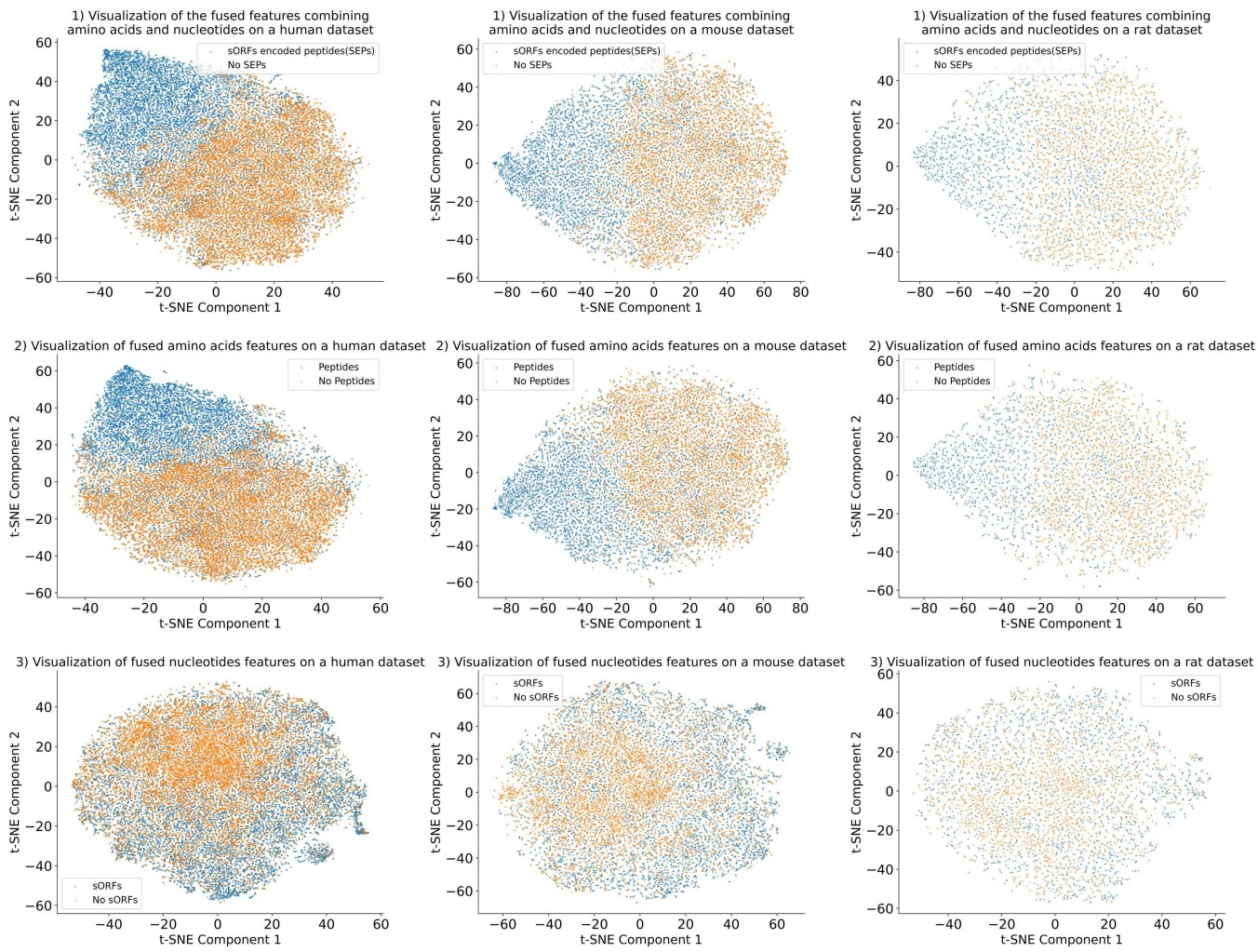

**Fig 5. Distribution of the AN feature, AA feature and NT feature on three different datasets.**

SHAP values of the top 20 features through a bar chart. The length of each bar represented the absolute magnitude of the average SHAP value of the corresponding feature, which intuitively reflected the average impact of each feature on the model output.

We selected features by taking the absolute average of SHAP values and tried other different thresholds. The results showed that the top 20 features significantly impact model performance, but other features also play important roles in specific situations. These results highlight the necessity of considering different types of features comprehensively rather than relying solely on a single feature encoding method. The combination of Amino Acids and Nucleotides features provided a wider range of information for protein sequence characterization than a single-feature encoding algorithm, thus demonstrating superior predictive performance.

The performance of four feature extraction methods was listed by seven evaluation metrics in Table 5. In detail, in the human dataset, the AN feature showed optimized performance in comparison with the AA Feature, NT Feature and ESM Feature, with improvement of ACC by 2.8%–3.7%, AUC by 5.1%–5.9%, MCC by 4.8%–6.9% and F1 by 1.4%–5.1%. In the mouse dataset, the AN feature exhibited superiority in comparison with the other three feature

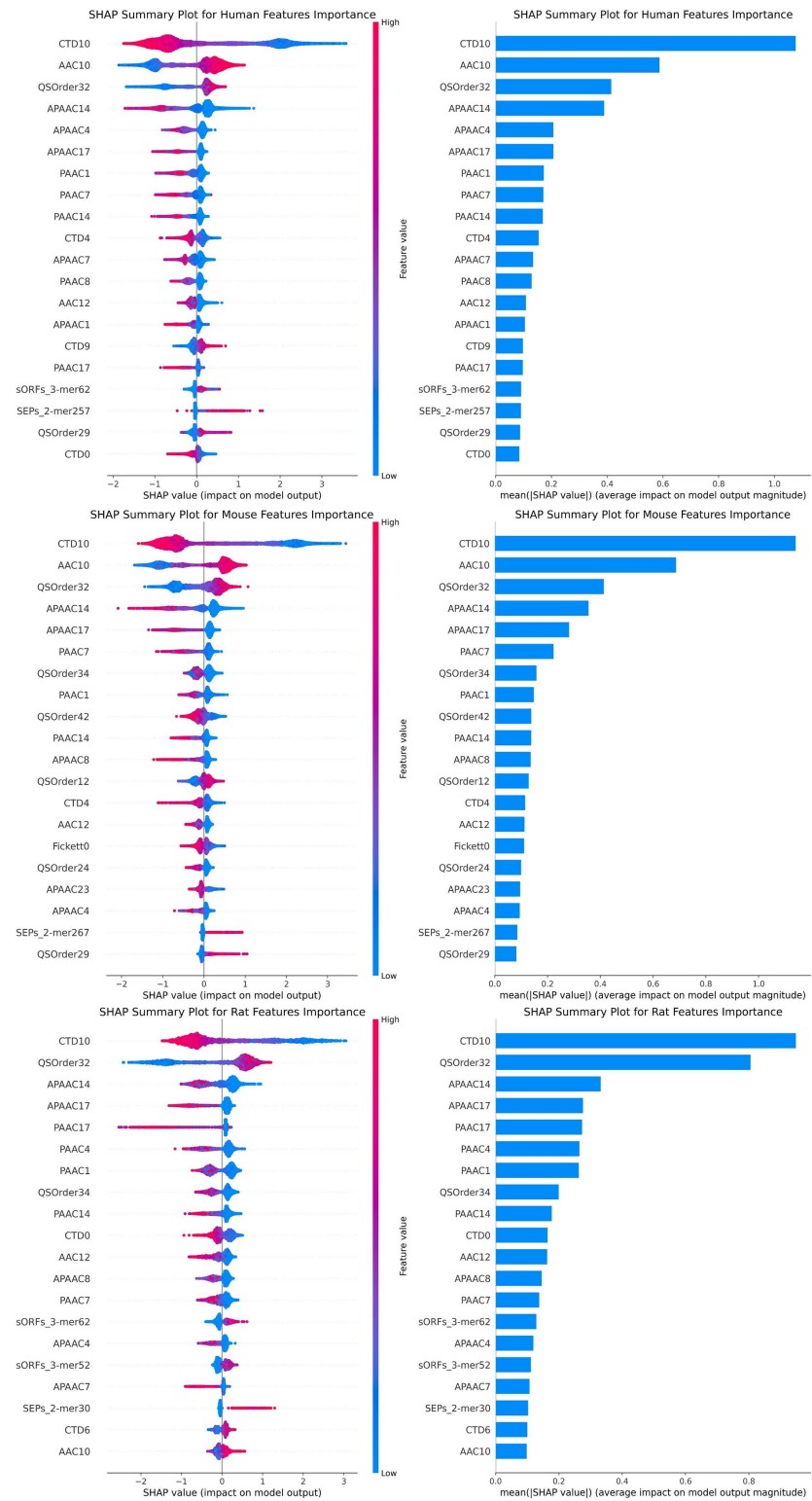

**Fig 6. Important features of SHAP interpretation. left column: The SHAP value distribution of the 20 most important features. right column: The importance bar chart of the top 20 features.**

**Table 5. Performance of AN Feature, AA Feature, NT Feature and ESM Feature on three species datasets.**

| Species | Feature | ACC | PRE | AUC | MCC | F1 | SN | SP |
|---|---|---|---|---|---|---|---|---|
| Human | **AN Features** | **0.839** | **0.853** | **0.933** | **0.679** | **0.837** | **0.822** | **0.857** |
| | AA Features | 0.811 | 0.766 | 0.882 | 0.631 | 0.823 | 0.889 | 0.736 |
| | ESM | 0.807 | 0.888 | 0.869 | 0.628 | 0.787 | 0.736 | 0.870 |
| | NT Features | 0.802 | 0.852 | 0.874 | 0.610 | 0.786 | 0.730 | 0.873 |
| Mouse | **AN Features** | **0.837** | **0.820** | **0.926** | **0.674** | **0.840** | **0.860** | **0.813** |
| | AA Features | 0.807 | 0.758 | 0.885 | 0.626 | 0.829 | 0.914 | 0.695 |
| | ESM | 0.805 | 0.828 | 0.868 | 0.611 | 0.794 | 0.763 | 0.800 |
| | NT Features | 0.787 | 0.832 | 0.847 | 0.578 | 0.763 | 0.704 | 0.865 |
| Rat | **AN Features** | **0.830** | **0.820** | **0.909** | **0.659** | **0.824** | **0.828** | **0.831** |
| | AA Features | 0.780 | 0.717 | 0.851 | 0.584 | 0.807 | 0.922 | 0.638 |
| | ESM | 0.758 | 0.746 | 0.816 | 0.516 | 0.750 | 0.746 | 0.756 |
| | NT Features | 0.732 | 0.806 | 0.810 | 0.480 | 0.700 | 0.619 | 0.848 |

Note: Amino Acids-Nucleotides fused feature (AN Feature), Amino acid fused feature (AA Feature), Nucleotide fused feature (NT Feature)

methods, with enhancement of ACC by 3%–5%, AUC by 4.1%–7.9%, MCC by 4.8%–9.6% and F1 by 1.1%–7.7%. In the rat dataset, the AN feature demonstrated higher evaluation metrics in comparison with the other three methods, with advancement of ACC by 5%–9.8%, AUC by 5.8%–9.9%, MCC by 7.5%–17.9% and F1 by 1.7%–12.4%.

Moreover, the performance of ESM Feature was demonstrated to be better than NT features by comparing the evaluation metrics results in Table 5. ESM Feature showed relatively minor differences in various evaluation metrics with comparison of AA Feature. But the precision metric of ESM Feature exhibited superiority in comparison with AA Feature. Therefore, the ESM-2 model was chosen to mine the prior knowledge representational information of SEPs.

## 3.2 Model comparison from different perspectives

The AN Feature and ESM Feature were selected after the feature comparison section. The sequence composition calculation and k-mer feature encoding methods were employed in the AN Feature, which caused some sequences to have feature values of 0 leading to sparse matrices. Preprocessing methods needed to be adopted to input sparse matrix features for input of deep learning model, such as L1 regularization (Lasso) [51], Sparse matrix-vector multiplication (SpMV) [52] and Sparse Matrix Factorization [53]. But the selected feature may have divergent impacts due to discrepancies in preprocessing methods. However, the ESM Feature showed good performance in section 3.1. ESM-2 was employed to overcome the challenges of processing sparse features in deep learning models. Therefore, machine learning models were utilized to process AN Feature.

A corresponding hidden vector for a given protein sequence was generated by using ESM-2 to represent protein sequence structural and functional features. Therefore, ESM-2 model offered more in-depth information of SEPs. Due to the intricate correlations within the ESM Feature, deep learning models were employed to capture the prior knowledge representational information of SEPs.

Therefore, model comparison was conducted from multi-perspective methods. This section will detail the methods and results of model selection.

**3.2.1 Comparison of machine learning models.** For the performance of machine learning models with AN Feature, eight individual machine learning models were compared,

including CatBoost, Random Forest (RF), LightGBM (LGBM), GBDT, Logistic Regression (LR), XGBoost, SVM and Decision Tree (DT). Through validation on three datasets, three models demonstrated excellent, including CatBoost, Random Forest and GBDT. And then, the stacking stratrgy was employed to fuse three models. These stacking models included combination Random Forest, CatBoost and GBDT (RCG), blend GBDT and CatBoost (GC), compound Random Forest and CatBoost (RC); and integration Random Forest and GBDT(RG). The specific comparative results can be found in Fig 7 and Table 6.

The red bars in the histogram of Fig 7 represented the CatBoost model and showed better performance in terms of ACC and MCC evaluation metrics. The blue curves in the ROC curve of Fig 7 symbolized the area under the curve (AUC) of the CatBoost model. The AUC of the CatBoost model was slightly lower than the GBDT model on the human AN Feature by 0.01% and on the mouse AN Feature by 0.15%. But the CatBoost model had shown superiority in comparison with other individual models and the four stacked models. Besides, the CatBoost model with AN Feature performed get best performance on the rat dataset.

The comparison results of machine learning models were presented in Table 6. Specifically, for the human AN Feature, the CatBoost model showed better performance on three evaluation metrics in comparison with other models, with advancement of ACC by 0.1%–4%, MCC by 0.3%–7.7% and F1 score by 0.1%–4.9%. For the mouse AN Feature, the CatBoost model exhibited superiority on three evaluation metrics in comparison with other models, with improvement of ACC by 0.1%–4.9%, MCC by 0.1%–9.7% and Precision by 0.2%–5.1%. For the rat AN Feature, the CatBoost model shown optimized performance on four evaluation metrics in comparison with other models, with enhancement of ACC by 0.5%–7.9%, MCC by 0.3%–15.9%, AUC by 0.5%–8.7% and F1 score by 0.7%–8.3%. The CatBoost model had been proven to have outstanding performance by comparing the evaluation metrics within three datasets. Therefore, the CatBoost model was chosen to process the AN Feature.

**3.2.2 Comparison of deep learning models.** For the performance of deep learning models with ESM Feature, 11 individual deep learning models were compared, including LSTM, GRU, RNN, CNN, MLP, TextCNN, BiLSTM, Self-attention (Attention), Attention_BiLSTM (ABiLSTM), Attention_BiGRU (ABiGRU) and CNN_LSTM (CLSTM). Among them, three fusion networks were detailed in Supporting information (S1 Table and S1-S3 Figs), such as ABiLSTM, ABiGRU and CLSTM. Three deep learning models performed well on human ESM Feature, like Attention, LSTM and GRU. And then, the stacking strategy was employed to combine three models. These stacking models contained fusion Attention, LSTM and GRU (ALG), compound Attention and LSTM (AL), integration Attention and GRU (AG) and combination LSTM and GRU (LG). Moreover, three deep learning models showed superiority with mouse ESM Feature and rat ESM Feature, including Attention, CNN and LSTM. Subsequently, the stacking strategy was utilized to blend three models. These stacking models incorporated mergence Attention, CNN and LSTM (ACL), composition Attention and CNN (AC), fusion Attention and LSTM (AL) and concoction CNN and LSTM (CL). The specific comparative results can be found in Fig 8 and Table 7.

The red bar in the histogram of Fig 8 symbolized the Attention model and displayed superiority in terms of ACC and MCC evaluation metrics. The blue curve in the ROC curve of Fig 8 represented the area under the curve (AUC) of the Attention model. The Attention model had exhibited better performance in comparison with other individual models and the four stacked models on all ESM Features.

The comparison results of deep learning models were shown in Table 7. Particularly, on the human ESM Feature, the Attention model exhibited superiority on three evaluation metrics in comparison with other models, with improvement of ACC by 0.1%–7.3%, MCC by 0.4%–9.6% and AUC by 0.2%–12%. On the mouse ESM Feature, the Attention model bespoke better

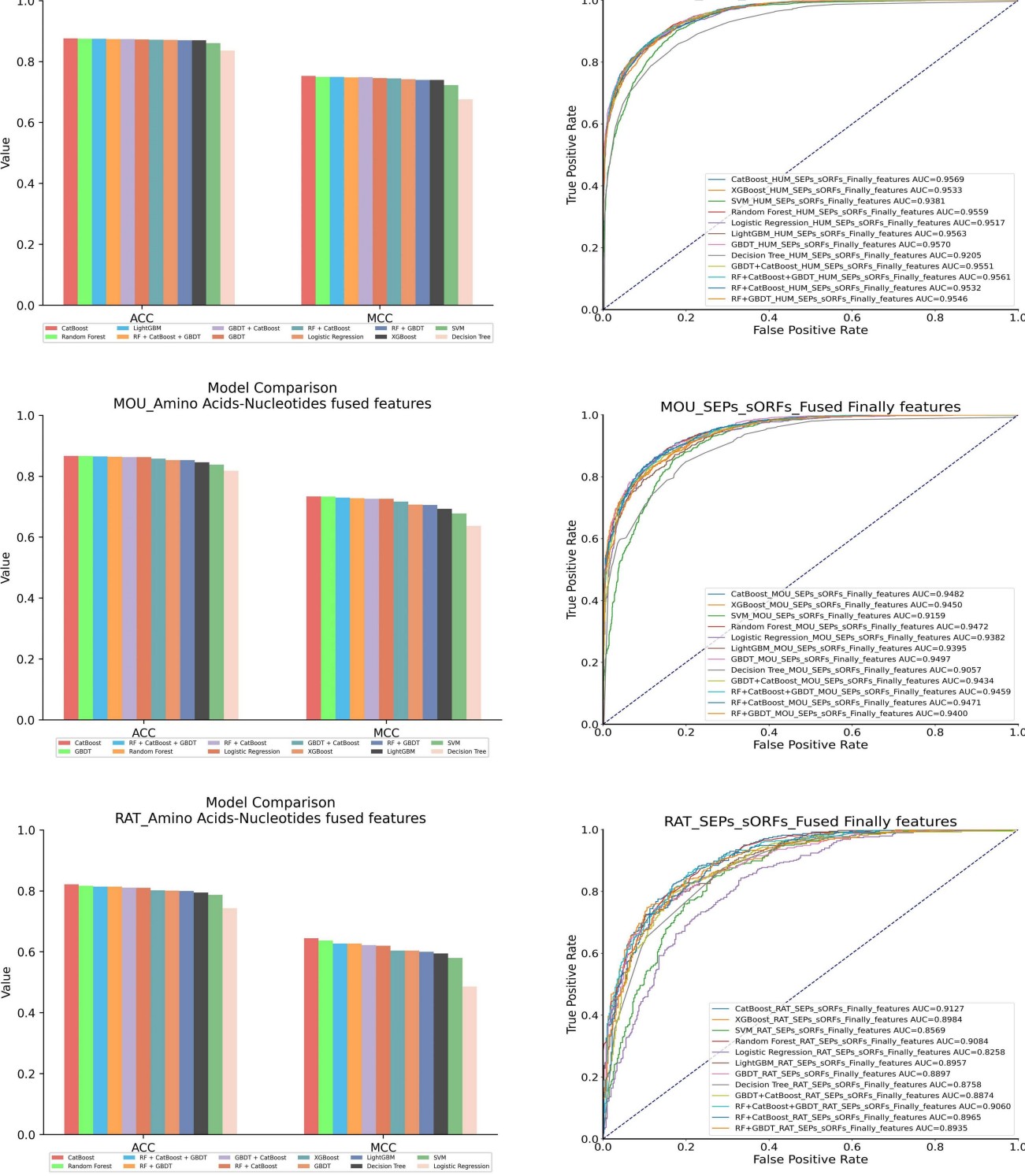

**Fig 7. Comparison of traditional encoding methods in machine learning models.**

**Table 6. Performance of machine learning models.**

| Feature | Model | ACC | PRE | AUC | MCC | F1 | SN | SP |
|---|---|---|---|---|---|---|---|---|
| AN Feature (Human) | **CatBoost** | **0.876** | **0.875** | **0.957** | **0.753** | **0.878** | **0.881** | **0.872** |
| | RF | 0.875 | 0.879 | 0.956 | 0.750 | 0.876 | 0.873 | 0.877 |
| | LGBM | 0.875 | 0.872 | 0.956 | 0.750 | 0.877 | 0.882 | 0.868 |
| | RCG | 0.874 | 0.878 | 0.956 | 0.748 | 0.874 | 0.870 | 0.878 |
| | GC | 0.874 | 0.879 | 0.955 | 0.749 | 0.874 | 0.870 | 0.879 |
| | GBDT | 0.873 | 0.870 | 0.957 | 0.746 | 0.873 | 0.875 | 0.870 |
| | RC | 0.872 | 0.881 | 0.953 | 0.745 | 0.872 | 0.862 | 0.883 |
| | LR | 0.871 | 0.865 | 0.952 | 0.742 | 0.875 | 0.885 | 0.857 |
| | RG | 0.870 | 0.878 | 0.955 | 0.740 | 0.869 | 0.861 | 0.879 |
| | XGBoost | 0.870 | 0.870 | 0.953 | 0.740 | 0.871 | 0.871 | 0.869 |
| | SVM | 0.861 | 0.837 | 0.938 | 0.723 | 0.866 | 0.896 | 0.825 |
| | DT | 0.836 | 0.874 | 0.920 | 0.676 | 0.829 | 0.788 | 0.885 |
| AN Feature (Mouse) | **CatBoost** | **0.867** | **0.859** | **0.948** | **0.734** | **0.864** | **0.869** | **0.865** |
| | GBDT | 0.866 | 0.857 | 0.950 | 0.733 | 0.867 | 0.877 | 0.856 |
| | RCG | 0.865 | 0.853 | 0.946 | 0.730 | 0.865 | 0.877 | 0.853 |
| | RF | 0.864 | 0.858 | 0.947 | 0.728 | 0.863 | 0.867 | 0.861 |
| | RC | 0.863 | 0.853 | 0.947 | 0.726 | 0.863 | 0.873 | 0.854 |
| | LR | 0.863 | 0.858 | 0.938 | 0.726 | 0.862 | 0.865 | 0.861 |
| | GC | 0.858 | 0.844 | 0.943 | 0.717 | 0.859 | 0.874 | 0.843 |
| | XGBoost | 0.853 | 0.846 | 0.945 | 0.707 | 0.851 | 0.855 | 0.852 |
| | RG | 0.853 | 0.842 | 0.940 | 0.706 | 0.853 | 0.863 | 0.843 |
| | LGBM | 0.846 | 0.836 | 0.939 | 0.693 | 0.848 | 0.860 | 0.832 |
| | SVM | 0.838 | 0.808 | 0.916 | 0.678 | 0.843 | 0.882 | 0.794 |
| | DT | 0.818 | 0.827 | 0.906 | 0.637 | 0.818 | 0.809 | 0.829 |
| AN Feature (Rat) | **CatBoost** | **0.822** | **0.802** | **0.913** | **0.645** | **0.818** | **0.836** | **0.810** |
| | RF | 0.817 | 0.790 | 0.908 | 0.637 | 0.818 | 0.847 | 0.789 |
| | RCG | 0.814 | 0.802 | 0.906 | 0.627 | 0.811 | 0.820 | 0.807 |
| | RG | 0.814 | 0.803 | 0.893 | 0.627 | 0.810 | 0.817 | 0.810 |
| | GC | 0.811 | 0.799 | 0.887 | 0.622 | 0.808 | 0.817 | 0.805 |
| | RC | 0.810 | 0.791 | 0.896 | 0.620 | 0.809 | 0.828 | 0.793 |
| | XGBoost | 0.802 | 0.804 | 0.898 | 0.604 | 0.797 | 0.791 | 0.813 |
| | GBDT | 0.801 | 0.775 | 0.890 | 0.604 | 0.804 | 0.836 | 0.767 |
| | LGBM | 0.800 | 0.768 | 0.896 | 0.600 | 0.792 | 0.818 | 0.784 |
| | DT | 0.795 | 0.760 | 0.876 | 0.595 | 0.813 | 0.875 | 0.711 |
| | SVM | 0.787 | 0.753 | 0.857 | 0.58 | 0.798 | 0.848 | 0.728 |
| | LR | 0.743 | 0.728 | 0.826 | 0.486 | 0.735 | 0.742 | 0.745 |

Note: Amino Acids-Nucleotides fused feature (AN Feature), Random Forest (RF), LightGBM (LGBM), Logistic-Regression (LR), Decision Tree (DT), RF + CatBoost + GBDT (RCG), GBDT + CatBoost (GC), RF + CatBoost (RC), RF + GBDT (RG)

performance on three evaluation metrics in comparison with other models, with enhancement of ACC by 0.7%–6.1%, MCC by 1%–11.4% and AUC by 0.8%–8.8%. On the rat ESM Feature, the Attention model emerged optimized performance on three evaluation metrics in comparison with other models, with advancement of ACC by 1.2%–8.5%%, MCC by 2.8%–17.3% and AUC by 1.7%–9.3%. The Attention model had demonstrated outstanding performance by comparing the evaluation metrics within three datasets. Therefore, the Attention model was selected to deal the ESM Feature.

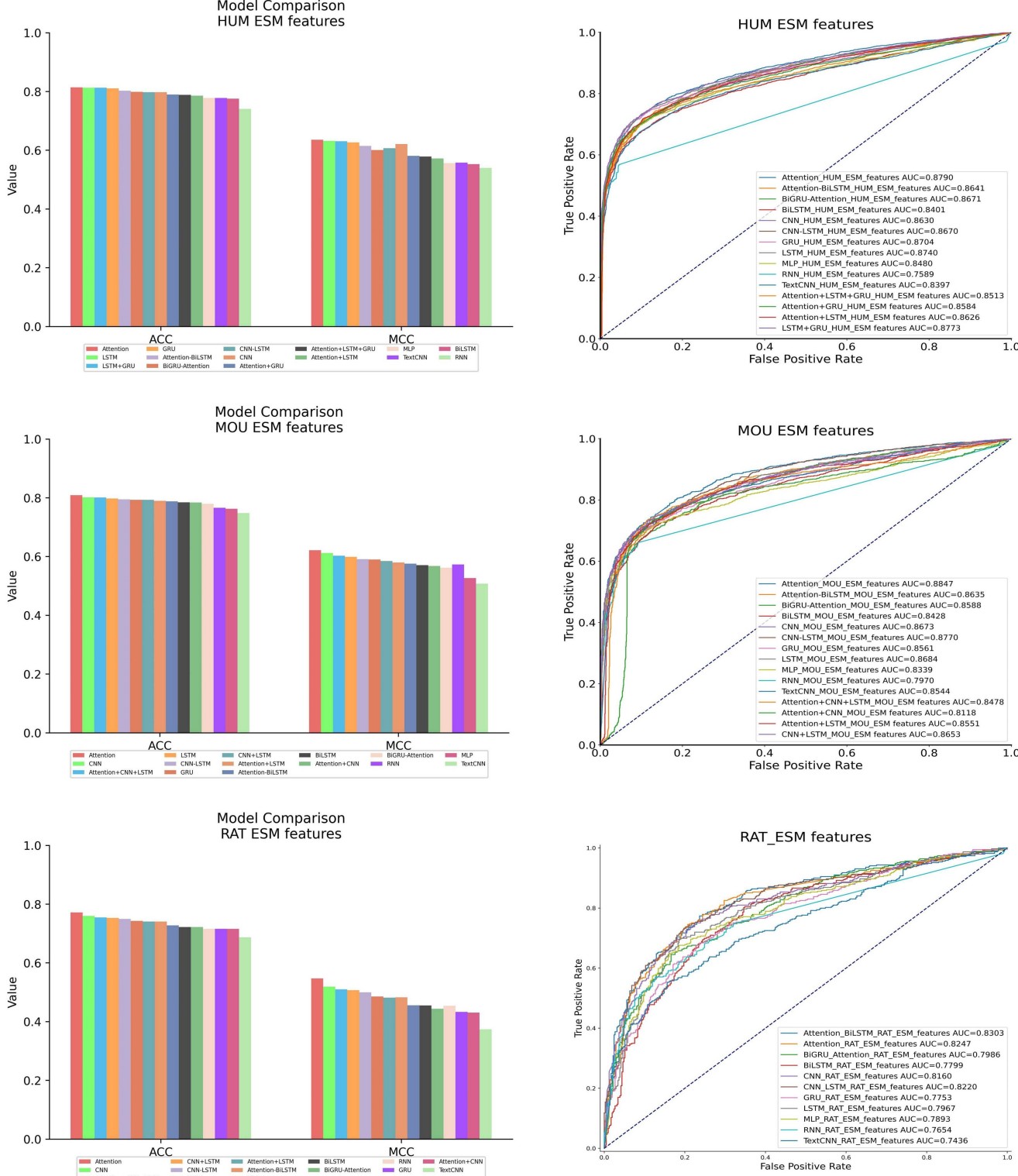

**Fig 8. Comparison of deep learning models.**

**Table 7. Performance of deep learning models.**

| Features | Model | ACC | PRE | AUC | MCC | F1 | SN | SP |
|---|---|---|---|---|---|---|---|---|
| ESM (Human) | **Attention** | **0.814** | **0.878** | **0.879** | **0.636** | **0.798** | **0.731** | **0.897** |
| | LSTM | 0.813 | 0.858 | 0.874 | 0.632 | 0.802 | 0.753 | 0.874 |
| | LG | 0.813 | 0.860 | 0.877 | 0.631 | 0.802 | 0.751 | 0.876 |
| | GRU | 0.811 | 0.857 | 0.870 | 0.627 | 0.799 | 0.749 | 0.874 |
| | ABiLSTM | 0.803 | 0.862 | 0.864 | 0.615 | 0.788 | 0.725 | 0.883 |
| | ABiGRU | 0.799 | 0.836 | 0.867 | 0.601 | 0.789 | 0.746 | 0.852 |
| | CLSTM | 0.798 | 0.864 | 0.867 | 0.607 | 0.780 | 0.710 | 0.887 |
| | CNN | 0.798 | 0.913 | 0.863 | 0.621 | 0.767 | 0.662 | 0.936 |
| | AG | 0.790 | 0.810 | 0.849 | 0.581 | 0.785 | 0.762 | 0.819 |
| | ALG | 0.789 | 0.806 | 0.855 | 0.579 | 0.785 | 0.766 | 0.813 |
| | AL | 0.786 | 0.799 | 0.845 | 0.572 | 0.782 | 0.766 | 0.805 |
| | MLP | 0.778 | 0.787 | 0.848 | 0.556 | 0.776 | 0.766 | 0.790 |
| | TextCNN | 0.778 | 0.799 | 0.840 | 0.558 | 0.772 | 0.747 | 0.810 |
| | BiLSTM | 0.776 | 0.797 | 0.840 | 0.553 | 0.770 | 0.745 | 0.808 |
| | RNN | 0.741 | 0.937 | 0.759 | 0.540 | 0.669 | 0.520 | 0.964 |
| ESM (Mouse) | **Attention** | **0.809** | **0.854** | **0.885** | **0.622** | **0.792** | **0.739** | **0.877** |
| | CNN | 0.802 | 0.861 | 0.867 | 0.612 | 0.780 | 0.713 | 0.888 |
| | ACL | 0.801 | 0.812 | 0.848 | 0.603 | 0.794 | 0.777 | 0.825 |
| | LSTM | 0.798 | 0.827 | 0.868 | 0.599 | 0.785 | 0.748 | 0.847 |
| | CLSTM | 0.795 | 0.817 | 0.877 | 0.591 | 0.783 | 0.751 | 0.837 |
| | GRU | 0.793 | 0.836 | 0.856 | 0.590 | 0.774 | 0.722 | 0.862 |
| | CL | 0.793 | 0.798 | 0.865 | 0.585 | 0.787 | 0.776 | 0.809 |
| | AL | 0.790 | 0.797 | 0.855 | 0.580 | 0.784 | 0.771 | 0.809 |
| | ABiLSTM | 0.788 | 0.779 | 0.864 | 0.576 | 0.787 | 0.796 | 0.781 |
| | BiLSTM | 0.785 | 0.810 | 0.843 | 0.571 | 0.771 | 0.736 | 0.832 |
| | AC | 0.784 | 0.781 | 0.812 | 0.568 | 0.781 | 0.782 | 0.786 |
| | ABiGRU | 0.780 | 0.797 | 0.859 | 0.562 | 0.769 | 0.743 | 0.816 |
| | RNN | 0.766 | 0.927 | 0.797 | 0.573 | 0.707 | 0.571 | 0.956 |
| | MLP | 0.763 | 0.758 | 0.834 | 0.527 | 0.761 | 0.765 | 0.762 |
| | TextCNN | 0.748 | 0.705 | 0.854 | 0.508 | 0.768 | 0.842 | 0.657 |
| ESM (Rat) | **Attention** | **0.772** | **0.807** | **0.837** | **0.547** | **0.749** | **0.699** | **0.841** |
| | CNN | 0.760 | 0.764 | 0.816 | 0.519 | 0.748 | 0.733 | 0.785 |
| | ACL | 0.755 | 0.737 | 0.794 | 0.510 | 0.754 | 0.771 | 0.739 |
| | CL | 0.753 | 0.739 | 0.819 | 0.507 | 0.751 | 0.763 | 0.744 |
| | CLSTM | 0.750 | 0.732 | 0.815 | 0.500 | 0.749 | 0.766 | 0.734 |
| | LSTM | 0.743 | 0.745 | 0.797 | 0.486 | 0.732 | 0.720 | 0.766 |
| | AL | 0.741 | 0.728 | 0.779 | 0.482 | 0.738 | 0.748 | 0.734 |
| | ABiLSTM | 0.741 | 0.720 | 0.820 | 0.483 | 0.742 | 0.766 | 0.717 |
| | MLP | 0.728 | 0.728 | 0.789 | 0.456 | 0.717 | 0.707 | 0.749 |
| | BiLSTM | 0.722 | 0.679 | 0.780 | 0.455 | 0.741 | 0.815 | 0.634 |
| | ABiGRU | 0.721 | 0.724 | 0.799 | 0.441 | 0.706 | 0.689 | 0.751 |
| | RNN | 0.716 | 0.821 | 0.765 | 0.454 | 0.646 | 0.532 | 0.890 |
| | GRU | 0.716 | 0.697 | 0.775 | 0.433 | 0.717 | 0.738 | 0.695 |
| | AC | 0.716 | 0.724 | 0.804 | 0.431 | 0.698 | 0.674 | 0.756 |
| | TextCNN | 0.687 | 0.697 | 0.744 | 0.374 | 0.663 | 0.632 | 0.739 |

Note: Self-attention (Attention), Attention_BiLSTM (ABiLSTM); Attention_BiGRU (ABiGRU); CNN_LSTM (CLSTM), LSTM+GRU (LG); Attention+GRU (AG); Attention+LSTM (AL); Attention+LSTM+GRU (ALG); CNN+LSTM (CL); Attention+CNN (AC); Attention+CNN+LSTM (ACL)

## 3.3 Performance comparison with state-of-the-art predictors

The performance of the SORFPP was validated by comparing with state-of-the-art methods from the past five years on three different species datasets, including CPPred, Mipepid, CPE-SLDI, DeepCPP, sORFplnc and sORFpred. Here was a brief overview of the features and models of comparative methods: 1) for CPPred, to identify coding RNAs, the SVM model was employed to process the traditional fusion feature that incorporated six nucleotide features and three amino acid features. 2) for Mipepid, to predict peptide, the Logistic Regression model was employed to deal the fusion feature that combined two nucleotide features. 3) for CPE-SLDI, to identify sORF, the oversampling was used to address the issue of class imbalance. 4) for DeepCPP: to predict coding RNAs, the CNN model was utilized to dispose the filtering feature that used the mDS feature selection method. 5) for sORFplnc, to identify sORF on plant imbalanced datasets, the resampling strategy was applied to handle the combination feature. 6) for sORFpred, to identify SEPs on plant datasets, the PCA dimensionality reduction was adopted to process the feature that extracted by the novel MCSEN model. The summary of these comparative methods was listed in Table 8.

The specific results of comparing SORFPP with other predictors were shown in Table 9. Furthermore, histograms, ROC and PR curves of Fig 9 were generated for a more comprehensive analysis of the SORFPP performance.

The red bar in the histogram of Fig 9 symbolized the SORFPP and showed outstanding performance in evaluation metrics, such as ACC, Precision (PRE), AUC, MCC and F1 score. ROC and PR curves were plotted in the rightmost two columns of Fig 9. The blue curve in the ROC curve of Fig 9 represented the area under the curve (AUC) of the SORFPP. Besides, the blue curve in the PR curve of Fig 9 represented the precision-recall ratio of the SORFPP. Observation of the curve in Fig 9 revealed that the ROC and PR curves of SORFPP wrapped other models. Therefore, the SORFPP demonstrated higher evaluation metrics in comparison with other predictors, including CPPred, MiPepid, CPE-SLDI, DeepCPP, sORFplnc and sORFpred.

State–of–the–art predictors were proven to yield different predictions on different datasets by the analysis of the results in Table 9. SORFPP exhibited optimal performance on various datasets. Specifically, on the human dataset, the SORFPP showed better performance on five evaluation metrics in comparison with other predictors, with enhancement of ACC by 6.6%–10.6%, PRE by 0.8%–9.7%, AUC by 7.8%–10.8%, MCC by 12.4%–21% and F1 score by 8.2%–11.6%. On the mouse dataset, the SORFPP projected superiority on five evaluation metrics in comparison with other predictors, with advancement of ACC by 6.4%–11.2%, PRE by 0.7%–9.7%, AUC by 6.5%–11.7%, MCC by 12.3%–22.3% and F1 score by 7.5%–11.7%. On the rat dataset, the SORFPP bespoke optimized performance on five evaluation metrics in comparison with other predictors, with improvement of ACC by 6.1%–12.4%, PRE by 4%–11%, AUC by 5.7%–13.3%, MCC by 12.2%–24.2% and F1 score by 6.9%–15.5%.

## 3.4 Comparison of SORFPP and sORFPred on three plant datasets

To better demonstrate the generalizability of SORFPP, this study collected three plant datasets from sORFPred, including Arabidopsis thaliana (Ath), Glycine max (Gmax), and Physcomitrella patens (Ppatens). The performance of SORFPP was compared with sORFPred using the same plant datasets. Detailed results were presented in Fig 10 and Table 10.

The red bar in the histogram of Fig 10 symbolized the SORFPP and showed outstanding performance in evaluation metrics, such as ACC, Precision (PRE), AUC, MCC and F1 score. ROC and PR curves were plotted in the rightmost two columns of Fig 10. The blue curve in the ROC curve of Fig 10 represented the area under the curve (AUC) of the SORFPP. Besides,

**Table 8. Overview of all state-of-the-art predictors.**

| Model | Year | Method | Features |
|---|---|---|---|
| CPPred | 2019 | The SVM model was employed to process the traditional fusion feature that selected by using the mRMR-IFS method | 1. CTD<br>2. ORF Length<br>3. ORF Coverage<br>4. Hexamer Score<br>5. Fickett Score<br>6. ORF Integrity<br>7. Protein-level encoding include Pl, Gravy, Instability |
| MiPepid | 2019 | The Logistic Regression model was employed to deal the fusion feature | 1. k-mer<br>2. Hexamer Score |
| CPE-SLDI | 2020 | The XGBoost model was used to process the fusion feature | 1. CTD<br>2. ORF Length<br>3. ORF Coverage<br>4. Hexamer Score<br>5. Fickett Score<br>6. ORF Integrity<br>7. Protein-level encoding include Pl, Gravy, Instability |
| DeepCPP | 2020 | A feature selection method named mDS was proposed. Besides, the CNN (Convolutional Neural Network) model was utilized to dispose the filtering feature that used the mDS feature selection method | 1. Mean Hexamer Score<br>2. ORF Coverage<br>3. Fickett Score<br>4. 2-gap&3-gap<br>5. Max ORF length<br>6. Nucleotide bias<br>7. 1-mer<br>8. 3-mer for Max ORF |
| sORFplnc | 2023 | A stacked model was formed by merging three machine learning models, including SVM, LGBM and LR. The stacking model was applied to process the fusion feature that selected by using resampling strategy | 1. 2*g-gap<br>2. codon bias<br>3. k-mer<br>4. sORFs length<br>5. Hexamer Score |
| sORFpred | 2023 | An automatic feature extraction model named MCSEN was introduce by combining multi-scale convolution and SENet to extract 512-dimensional feature. Furthermore, A stacked model was built by fusing two machine learning models, including Extra Trees and LR. The stacking model was employed to process the fusion feature that combined the MCSEN feature and traditional features | 1. k-mer<br>2. SN<br>3. sORF length<br>4. GC content&ratio<br>5. SSM<br>6. CTD<br>7. Fickett Score<br>8. Hexamer Score<br>9. Protein-level encoding includes AAC, GAAC, GTPC, CKSAAGP, CTD, GDPC, KSCTriad<br>10. MCSEN model |

the blue curve in the PR curve of Fig 10 represented the precision-recall ratio of the SORFPP. Observation of the curve in Fig 10 revealed that the ROC and PR curves of SORFPP wrapped sORFPred.

In Table 10, SORFPP demonstrated superior performance to sORFPred across all evaluation metrics on the same datasets. Specifically, on the three datasets, the SORFPP showed better performance on five evaluation metrics in comparison with sORFPred, with enhancement of ACC by 5.6%–6.4%, PRE by 6%–7.6%, AUC by 2.5%–3.9%, MCC by 9.9%–12.7% and F1 score by 5.1%–6.4%.

### 3.5 Distribution of the Predicted Score

The prediction scores indicated the probability that the sequence was an SEP. The heatmap of the prediction scores was used to visualize the predictive capabilities of SORFPP and other

**Table 9. Results of SORFPP and other predictors on different datasets.**

| Species | Method | ACC | PRE | AUC | MCC | F1 | SN | SP |
|---------|--------|-----|-----|-----|-----|-----|-----|-----|
| Human | **SORFPP** | **0.882** | **0.887** | **0.958** | **0.764** | **0.882** | **0.877** | **0.887** |
| | sORFpred | 0.816 | 0.879 | 0.880 | 0.640 | 0.800 | 0.735 | 0.898 |
| | sORFplnc | 0.798 | 0.839 | 0.869 | 0.600 | 0.786 | 0.740 | 0.856 |
| | CPE_SLDI | 0.798 | 0.842 | 0.862 | 0.602 | 0.786 | 0.737 | 0.860 |
| | CPPred | 0.787 | 0.841 | 0.859 | 0.581 | 0.766 | 0.703 | 0.870 |
| | DeepCPP | 0.780 | 0.814 | 0.855 | 0.563 | 0.769 | 0.729 | 0.832 |
| | MiPepid | 0.776 | 0.790 | 0.850 | 0.554 | 0.771 | 0.752 | 0.801 |
| Mouse | **SORFPP** | **0.875** | **0.858** | **0.946** | **0.750** | **0.875** | **0.893** | **0.857** |
| | sORFpred | 0.811 | 0.858 | 0.881 | 0.627 | 0.794 | 0.739 | 0.881 |
| | CPE_SLDI | 0.806 | 0.851 | 0.864 | 0.617 | 0.800 | 0.754 | 0.861 |
| | sORFplnc | 0.796 | 0.821 | 0.875 | 0.593 | 0.784 | 0.750 | 0.841 |
| | CPPred | 0.780 | 0.840 | 0.841 | 0.569 | 0.758 | 0.691 | 0.869 |
| | DeepCPP | 0.779 | 0.786 | 0.853 | 0.558 | 0.772 | 0.757 | 0.800 |
| | MiPepid | 0.763 | 0.761 | 0.829 | 0.527 | 0.761 | 0.761 | 0.766 |
| Rat | **SORFPP** | **0.847** | **0.845** | **0.922** | **0.694** | **0.843** | **0.841** | **0.854** |
| | sORFplnc | 0.786 | 0.796 | 0.848 | 0.572 | 0.774 | 0.753 | 0.817 |
| | CPE_SLDI | 0.781 | 0.784 | 0.865 | 0.562 | 0.772 | 0.762 | 0.800 |
| | sORFpred | 0.753 | 0.776 | 0.840 | 0.508 | 0.733 | 0.694 | 0.810 |
| | DeepCPP | 0.741 | 0.745 | 0.810 | 0.481 | 0.728 | 0.712 | 0.768 |
| | MiPepid | 0.726 | 0.735 | 0.794 | 0.452 | 0.717 | 0.700 | 0.751 |
| | CPPred | 0.723 | 0.805 | 0.789 | 0.465 | 0.688 | 0.601 | 0.850 |

predictors. Fig 11 illustrated that darker colors represented SEPs and lighter colors indicated non-SEPs. SORFPP predictor was presented in the second column from the right of Fig 11. The last column displayed the true labels of the data. Inspection of the heatmap in Fig 11 revealed that the SORFPP showed a more distinct differentiation.

### 3.6 Application of the SORFPP

For the convenience of subsequent researchers to predict SEP, the source codes of the SORFPP framework have been available on the GitHub platform. The SORFPP framework was built by using the Python programming language to ensure compatibility with multiple operating systems, such as Windows and Linux. The content is obtained at the following URL: https://github.com/OR2513/SORFPP. Additionally, a user-friendly graphical interface was developed to utilize SORFPP. The interface was made immediately accessible at http://111.229.198.94:5000/.

## 4. Discussion

Early research indicated that lncRNA sequences could not encode proteins. In recent years, the small open reading frame (sORF) in lncRNA sequence has been found to be capable of encoding peptide. This sORF-encoded peptide was known as SEP. SEP sequence had a lesser molecular weight in comparison with traditional peptide sequence, but it contained important functional structures, such as helices and folds. Besides, SEPs were frequently involved in biological processes, including cell signaling, intercellular interaction, gene expression control, and so on. However, the lack of a comprehensive lncRNA-encoded peptide database and the limitations of feature and model methods within previous predictors made predicting SEPs

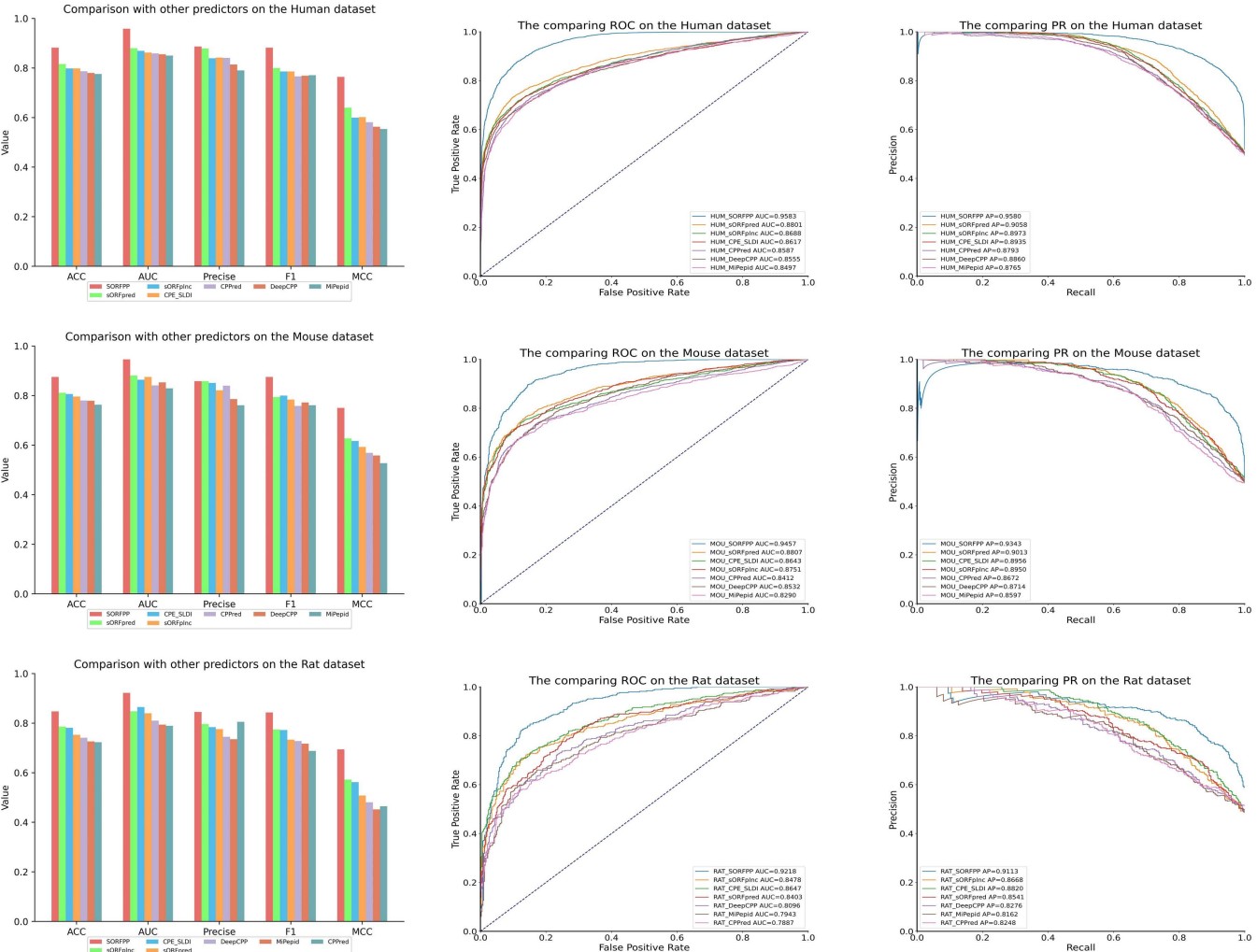

**Fig 9. Histogram, ROC curve and precision-recall curve plots of SORFPP and other predictors on three datasets.** The red bar in the histogram of Fig 9 symbolized the SORFPP. The blue curve in the ROC curve of Fig 9 represented the area under the curve (AUC) of the SORFPP.

challenging. In addition, previous research has focused on plant datasets. Therefore, SORFPP was proposed to improve the performance of predicting SEP by combining machine learning and deep learning methods on the experimentally validated dataset TransLnc.

The contribution of SORFPP to predicting SEPs was categorized into four aspects, including dataset, feature, model and web. 1) Dataset: SORFPP addressed the issues of species monotonicity and authenticity by collecting and processing the experimentally validated dataset TransLnc. 2) Feature: SORFPP extracted feature from multiple perspectives. Integrating nucleotide-level features into traditional amino acid features had a positive effect. Furthermore, ESM-2 could capture the prior knowledge representational information of SEPs. 3) The multi-perspective fusion model computational framework was utilized to address the issue of model monotonicity. Three models were integrated to calculate probability of predicting SEP sequences, such as CatBoost, self-attention and logistic regression. 4) A user-friendly graphical interface for SORFPP was developed to facilitate subsequent research.

Currently, the lack of specifically collated databases for prediction of SEPs was a significant issue. TransLnc validated many SEP and sORF sequence data by using various biological

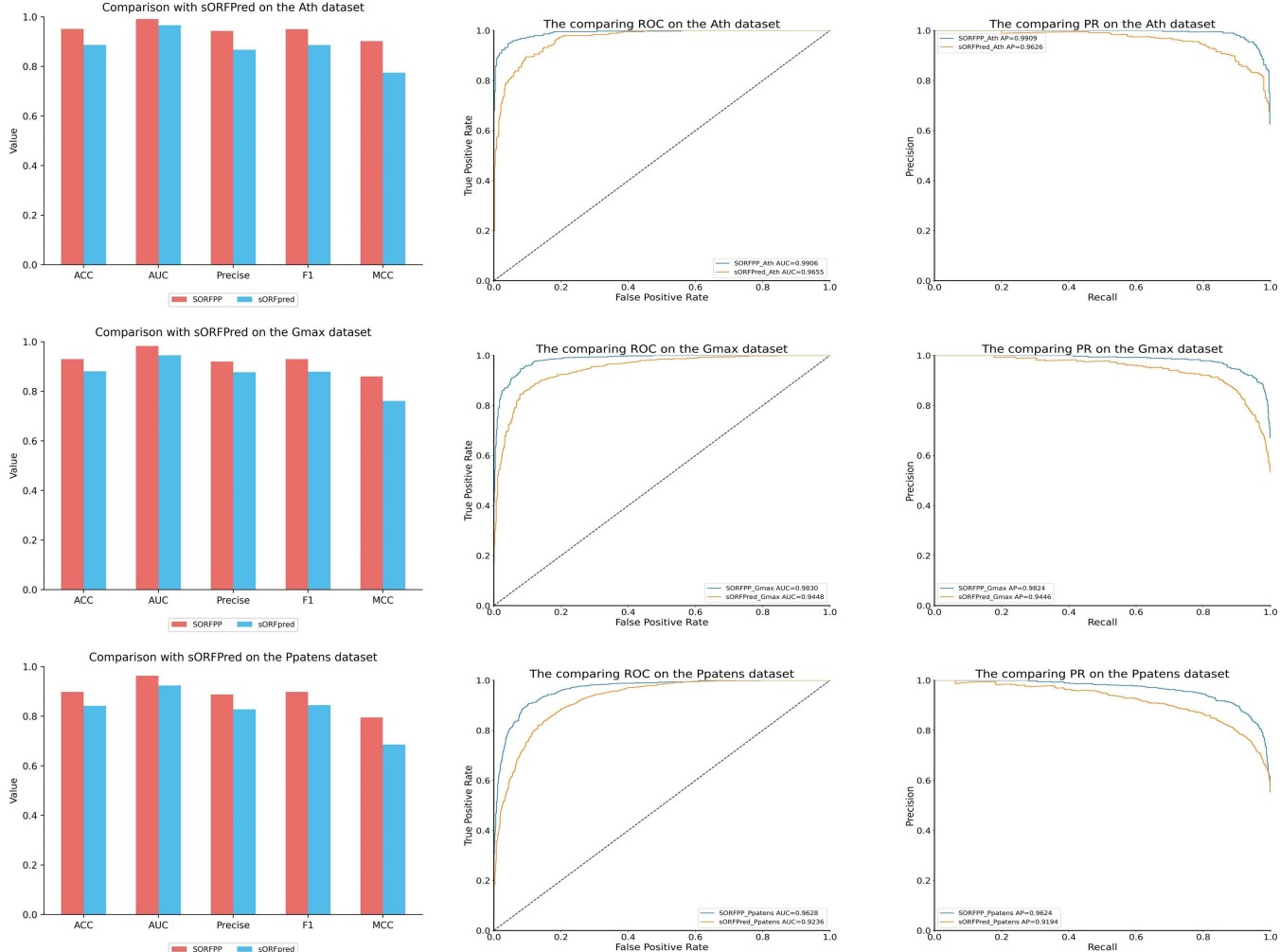

**Fig 10. Histogram, ROC curve and precision-recall curve plots of SORFPP and sORFPred on three plant datasets.**

Table 10. Results of SORFPP and sORFPred on three plant datasets.

| Species | Method | ACC | PRE | AUC | MCC | F1 | SN | SP |
|---|---|---|---|---|---|---|---|---|
| Ath | **SORFPP** | **0.951** | **0.943** | **0.991** | **0.902** | **0.95** | **0.957** | **0.945** |
| | sORFPred | 0.887 | 0.867 | 0.966 | 0.775 | 0.886 | 0.906 | 0.869 |
| Gmax | **SORFPP** | **0.93** | **0.92** | **0.983** | **0.86** | **0.93** | **0.921** | **0.936** |
| | sORFPred | 0.881 | 0.877 | 0.945 | 0.761 | 0.879 | 0.881 | 0.88 |
| Ppatens | **SORFPP** | **0.898** | **0.888** | **0.963** | **0.795** | **0.898** | **0.909** | **0.887** |
| | sORFPred | 0.842 | 0.828 | 0.924 | 0.686 | 0.845 | 0.862 | 0.824 |

Note: Arabidopsis thaliana (Ath); Glycine max (Gmax); Physcomitrella patens (Ppatens)

methods. Besides, TransLnc included three species datasets, such as human, mouse and rat. SORFPP obtained a reliable dataset by collecting and processing the TransLnc dataset.

Features play an important role in predicting SEPs. SEP sequences were lesser molecular weights in comparison to traditional peptide sequences. Therefore, a multi-perspective

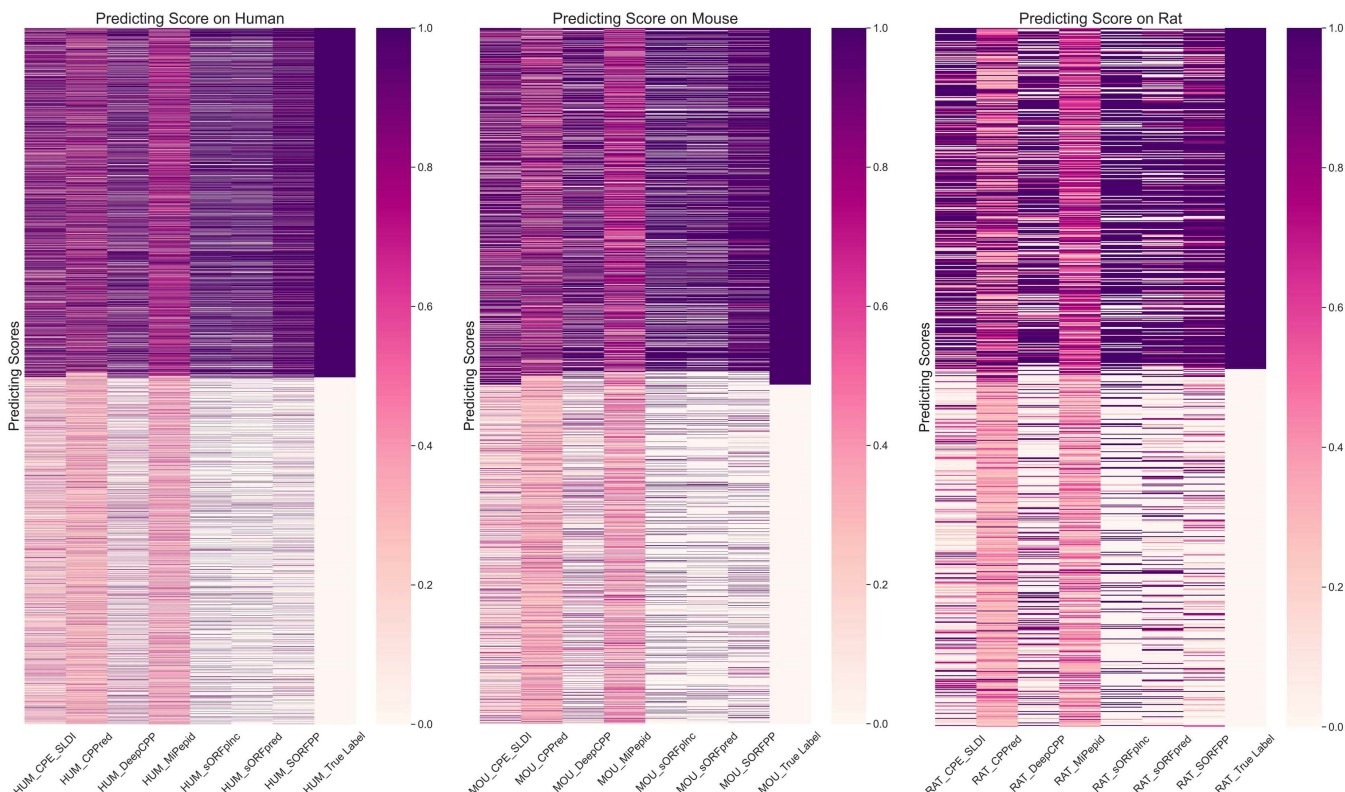

**Fig 11. The distribution of predicting score heatmaps on three datasets.** SORFPP predictor was presented in the second column. The last column displayed the true labels of the data.

feature extraction method was required to enrich SEP sequence information. 1) In traditional feature encoding methods, the sequence composition calculation and k-mer feature encoding methods of amino acids demonstrated better performance in comparison with other methods, including AAC, APAAC, PAAC, QSOrder, 2-mer and 4-mer. Similarly, the sequence composition calculation and k-mer feature encoding methods of nucleotides also exhibited superior performance in comparison with other methods, such as 3-mer, 4-mer, Fickett, and CTD. Therefore, the enrichment of sequence composition information has been shown to have a positive impact on the prediction of SEPs. And then, the ten feature encoding methods from amino acid and nucleotide were integrated to construct a combination feature (AN Feature). 2) Moreover, the protein language model of ESM-2 could generate a corresponding hidden vector for a given protein sequence for representing its structural and functional features. Thus, ESM-2 could capture the prior knowledge representational information of SEPs (ESM Feature). Besides, ESM-2 was employed to overcome the challenges of processing sparse features in deep learning models.

The models of the state-of-the-art methods did not exhibit satisfactory performance. Thus, SORFPP employed the multi-perspective fusion model computational framework. Due to the CatBoost model construction by using a balanced decision tree, the CatBoost model could process the missing and sparse data to capture non-zero features by splitting features. The self-attention mechanism enables the model to capture long-term dependencies between information in the SEPs and selectively assign higher weights to important SEPs sequence information and assign lower weights to other information. The logistic regression (LR) model as the final

decision-layer model utilized the linear combination feature. The LR model can find effective decision boundaries in effective feature spaces for the ESM Feature. Moreover, the LR model was capable of learning and utilizing collaborative relationships within the AN Feature.

Additionally, the performance of the SORFPP was validated by comparing with state-of-the-art methods from the past five years on three different species datasets, including CPPred, Mipepid, CPE-SLDI, DeepCPP, sORFplnc and sORFpred. The SORFPP showed better performance on five evaluation metrics in comparison with other predictors, with enhancement of ACC by 6.1%–12.4%, PRE by 4%–11%, AUC by 5.7%–13.3%, MCC by 12.2%–24.2%, and F1 score by 6.9%–15.5%. For the convenience of subsequent researchers to predict SEP, a user–friendly graphical interface was developed to utilize SORFPP. The interface was made immediately accessible at http://111.229.198.94:5000/.

## 5. Limitations and future works

SORFPP has shown better performance in predicting the biological activity of SEPs compared to other methods, but we believed this method still has certain limitations and there is room for advancement in the future.

Although the TransLnc dataset has compiled a wealth of sequence resources, sequencing technology was becoming more cost-effective, high-throughput, and versatile with its development. Fourth-generation sequencing technologies were maturing, including nanopore sequencing technology and electron microscopy, which can improve sequencing accuracy. In the future, with the construction of such databases becoming more comprehensive, we will continue to collect related datasets and supplement data samples.

Secondly, although the feature encoding methods for traditional peptide sequences were well-developed, research on encoding methods for this new type of SEP sequences was insufficient. The development of protein language models could better mine the sequence information of short peptides. For instance, newly introduced protein models like Ankh and the xTrimo Protein General Language Model (xTrimoPGLM) have demonstrated strong capabilities and broad prospects in mining hidden biological information within protein sequences. In future work, we will integrate the strengths of these models to more thoroughly mine the SEPs sequence information.

Most existing methods relied on comparative experiments to select the optimal machine learning or deep learning framework for building predictive models. Due to the vast number of machine learning and deep learning algorithms, this has become a time-consuming and labor-intensive process. Moreover, deep learning frameworks usually require significant computational resources and time to train and optimize models. To address these drawbacks, automated machine learning (AutoML) packages represented a deep neural network architecture that excels in identifying model performance based on the characteristics of the input information, such as Auto-PyTorch and AutoKeras. Additionally, these tools could help simplify the model optimization process in DL model training, greatly facilitating model training and enhancing the robustness of the trained models. In future work, we plan to integrate AutoML methods into the SEPs prediction process, significantly improving the predictive efficiency of the existing framework.

Looking ahead, we will concentrate on creating more robust models, researching more effective features and improving the predictive accuracy of SEPs.

### Highlights

1. **Dataset:** SORFPP has created a dependable experimental dataset by compiling and analyzing the TransLnc dataset, which utilizes various biological methods to validate numerous SEP and small open reading frame (sORF) sequences.

2. **Feature:** Due to the property of SEPs having a low molecular weight, SORFPP utilizes a multi-perspective method to extract SEP sequences information. The enrichment of sequence composition information shows positive impact on the prediction of SEPs and ESM-2 could capture the prior knowledge representational information of SEP sequences.

3. **Model:** The multi-perspective fusion model computational framework is utilized to enhance model performance. The CatBoost model can process the missing and sparse data to capture non-zero features by splitting features. The self-attention model can capture long-term dependencies between information in the SEPs and selectively assign weights to SEP sequences information. The LR model as the final decision-layer model utilizes the linear combination feature.

4. **Web:** For the convenience of subsequent researchers to predict SEP, this experiment develops a graphical user interface that directly uses SORFPP. Users can input sequences of SEP and sORF to obtain the probability of SEP activity.

## 6. Conclusion

SORFPP of a fused computational framework was presented to predict the activity of SEPs by integrating multi-perspective features and models on the validated experimental dataset TransLnc. SORFPP consists of three components: 1) The CatBoost model was employed to process the fusion feature that combined amino acids with nucleotides (AN Feature). 2) The self-attention model was utilized to manage the prior knowledge representation information of SEP sequence (ESM Feature) that extracted by the ESM-2 model. 3) The logistic regression model was adopted to calculate the fusion probability that combined the prediction results of the CatBoost model and the self-attention model.

To identify effective features for predicting SEP, feature extraction methods were selected from multiple perspectives. The sequence composition calculation and k-mer feature encoding methods have been proven to contribute positively to the prediction of SEPs through the comparison of features. Thus, the AN Feature was constituted by fusing ten traditional feature encoding methods from amino acid and nucleotide, including k-mer of amino acid and nucleotide, AAC, QSOrder, PAAC, APAAC, CTD and Fickett. To enrich the prior knowledge representation information of SEP sequences, the ESM-2 model was employed to extract the ESM Feature. Moreover, a multi-perspective method was chosen to build a comprehensive integrated model computational framework. The CatBoost model processed the AN Feature to effectively segment sparse values. The self-attention model was utilized to process the ESM Feature by selectively assigning appropriate weights to SEP sequence information. The logistic regression model was adopted to calculate the fusion probability that combined the prediction results of the CatBoost model and the self-attention model.

On three species datasets, SORFPP showed better performance on two evaluation metrics in comparison with state-of-the-art predictors, with enhancement of ACC by 6.1%-12.4% and MCC by 12.2%-24.2%. Additionally, the heatmap of prediction scores for SORFPP and six other methods also visually demonstrated that SORFPP was a reliable predictor for SEP.

## Supporting information

**S1 Table. Explanation of three fusion networks** .
(DOCX)

**S1 Fig. ABiLSTM Network.**
(TIF)

**S2 Fig. ABiGRU Network.**
(TIF)

**S3 Fig. CLSTM Network.**
(TIF)

## Author contributions

**Conceptualization:** Hongqi Feng, Sen Yang.

**Data curation:** Qi Nie.

**Formal analysis:** Sen Yang.

**Funding acquisition:** Hongqi Feng, Qi Nie, Sen Yang.

**Investigation:** Qi Nie.

**Methodology:** Qi Nie, Sen Yang.

**Project administration:** Hongqi Feng.

**Software:** Qi Nie.

**Validation:** Qi Nie.

**Visualization:** Qi Nie.

**Writing – original draft:** Qi Nie.

**Writing – review & editing:** Hongqi Feng, Qi Nie, Sen Yang.

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
