## [Decision Letter · Decision Letter 0]

10 Dec 2024

PONE-D-24-29982SORFPP: Enhancing Rich Sequence-driven Information to Identify SEPs Based on Fused Framework on Validation DatasetsPLOS ONE

Dear Dr. Nie,

Thank you for submitting your manuscript to PLOS ONE. After careful consideration, we feel that it has merit but does not fully meet PLOS ONE’s publication criteria as it currently stands. Therefore, we invite you to submit a revised version of the manuscript that addresses the points raised during the review process.

We look forward to receiving your revised manuscript.

Kind regards,

Prabina Kumar Meher, Ph.D.

Academic Editor

PLOS ONE

1. Y: Natural Science Foundation of Jiangsu Province of China (Grant No. BK20230626),

2. Y: State Key Laboratory of Plant Environmental Resilience (Grant No. SKLPERKF2401), 

3. Y: State Key Laboratory of Animal Biotech Breeding (Grant No. 2024SKLAB6-1), 

4. N: Postgraduate Research & Practice Innovation Program of Jiangsu Province (Grant No. KYCX23_3069),

5. Y: Fourth Batch of Leading Innovative Talents Introduction and Training Projects under the Longcheng Talent Plan in Changzhou City (Basic Research and Innovation) (Grant No. CQ20230086). 

4. We note you have included a table to which you do not refer in the text of your manuscript. Please ensure that you refer to Table 2 in your text; if accepted, production will need this reference to link the reader to the Table.

Reviewers' comments:

**Comments to the Author**

1. Is the manuscript technically sound, and do the data support the conclusions?

Reviewer #1: Partly

Reviewer #2: Yes

2. Has the statistical analysis been performed appropriately and rigorously? 

Reviewer #1: I Don't Know

Reviewer #2: Yes

3. Have the authors made all data underlying the findings in their manuscript fully available?

Reviewer #1: No

Reviewer #2: Yes

4. Is the manuscript presented in an intelligible fashion and written in standard English?

Reviewer #1: Yes

Reviewer #2: Yes

5. Review Comments to the Author

Reviewer #1: 1. While the paper introduces a new computational method, SORFPP, for predicting short open reading frames-encoded peptides (SEPs), the objectives and unique contributions are not clearly distinguished from previous studies.

2. The manuscript compares SORFPP against other state-of-the-art models using three benchmark datasets. However, it would be beneficial to include statistical significance testing to determine whether the observed performance improvements are genuinely significant.

3. The study employs multiple feature extraction methods, combining traditional and deep learning features. However, there is limited discussion on how feature selection was conducted.

4. Although the SORFPP model outperforms others in terms of prediction accuracy, the interpretability of the ensemble learning framework remains unclear. Explain how the model interprets or ranks feature importance, especially since SEPs are biologically significant.

5. The study uses datasets from human, mouse, and rat species for training and evaluation. However, the paper should address how well SORFPP generalizes to other species or datasets not included in this analysis.

6. Machine learning is well-known and has been used in previous sequence analysis studies i.e., PMID: 37649385, PMID: 36174933. Therefore, the authors are suggested to refer to more works in this description to attract a broader readership.

7. While the paper mentions limitations of alignment-based methods, a direct comparison with such methods on the same datasets would offer valuable insights into the advantages and disadvantages of using SORFPP.

8. The manuscript does not discuss the computational requirements or efficiency of the SORFPP model, which is crucial for high-throughput applications.

9. While the manuscript uses standard metrics such as accuracy, F1 score, and AUC, discussing the implications of these results in real-world scenarios, such as predicting SEPs in large-scale genomic projects, would be beneficial.

10. The manuscript would be strengthened by explicitly discussing the limitations of SORFPP and suggesting future research directions.

11. Some figures and tables could benefit from more detailed captions and explanations.

Reviewer #2: The manuscript develops a method called SORFPP, to predict SEPs by combining deep learning and machine learning together with protein representations. Developing web-based tool is very nice. The manuscript is written quite well, but there are weaknesses that need to be handled:

1- Can you give more details about TransLnc dataset?

2- What does evidence column mean in Table 1?

3- Is original dataset in Table 2 balanced? Or did the authors made it balanced?

4- In Algorithm 1, is indentation correct? Especially second for loop

Syntax errors:

1- In Page 25, thro ugh -> through

2- In Page 28, enchancemen -> enhancement

3- In Page 21, compriseed -> comprised

6. PLOS authors have the option to publish the peer review history of their article (what does this mean? ). If published, this will include your full peer review and any attached files.

**Do you want your identity to be public for this peer review?** For information about this choice, including consent withdrawal, please see our Privacy Policy .

Reviewer #1: No

Reviewer #2: No

---

## [Author Response · Author response to Decision Letter 1]

26 Dec 2024

Dear Editor and Reviewers:

Thank you for taking out of your busy schedule to review the manuscript. Now we have carefully revised and replied to the manuscript for this version. The revision instructions are as follows:

To Reviewer: 1

Question 1. While the paper introduces a new computational method, SORFPP, for predicting short open reading frames-encoded peptides (SEPs), the objectives and unique contributions are not clearly distinguished from previous studies.

Answer: Thanks for your suggestion. SEP sequence had a lesser molecular weight in comparison with traditional peptide sequence, but it contained important functional structures, such as helices and folds. Besides, SEPs were frequently involved in biological processes, including cell signaling, intercellular interaction, gene expression control, and so on. However, the lack of a comprehensive lncRNA-encoded peptide database and the limitations of feature and model methods within previous predictors made predicting SEPs challenging. In addition, previous research had focused on plant datasets. We had integrated a unique dataset named TransLnc, which encompassed a variety of species including humans, mice, and rats. The richness of this dataset was expected to enhance the depth and scope of our investigative work. Therefore, SORFPP was proposed to improve the performance of predicting SEP by combining machine learning and deep learning methods on the experimentally validated dataset TransLnc. The contribution of SORFPP was elaborated from four perspectives: dataset, feature extraction, model construction, and web interface. In the Highlights section of the paper, we provided a detailed discussion on these aspects.

Highlights

1. Dataset: SORFPP has created a dependable experimental dataset by compiling and analyzing the TransLnc dataset�which utilizes various biological methods to validate numerous SEP and small open reading frame (sORF) sequences.

2. Feature: Due to the property of SEPs having a low molecular weight, SORFPP utilizes a multi-perspective method to extract SEP sequences information. The enrichment of sequence composition information shows positive impact on the prediction of SEPs and ESM-2 could capture the prior knowledge representational information of SEP sequences.

3. Model: The multi-perspective fusion model computational framework is utilized to enhance model performance. The CatBoost model can process the missing and sparse data to capture non-zero features by splitting features. The self-attention model can capture long-term dependencies between information in the SEPs and selectively assign weights to SEP sequences information. The LR model as the final decision-layer model utilizes the linear combination feature.

4. Web: For the convenience of subsequent researchers to predict SEP, this experiment develops a graphical user interface that directly uses SORFPP. Users can input sequences of SEP and sORF to obtain the probability of SEP activity.

Question 2. The manuscript compares SORFPP against other state-of-the-art models using three benchmark datasets. However, it would be beneficial to include statistical significance testing to determine whether the observed performance improvements are genuinely significant.

Answer: Thanks for your suggestion. The prediction scores indicated the probability that the sequence was an SEP. The heatmap of the prediction scores was used to visualize the predictive capabilities of SORFPP and other predictors. Figure 11 illustrated that darker colors represented SEPs and lighter colors indicated non-SEPs. SORFPP predictor was presented in the second column from the right of Figure 11. The last column displayed the true labels of the data. Inspection of the heatmap in Figure 11 revealed that the SORFPP showed a more distinct differentiation. Refer to page 40 of the paper.

Question 3. The study employs multiple feature extraction methods, combining traditional and deep learning features. However, there is limited discussion on how feature selection was conducted.

Answer: Thanks for your suggestion. In this paper, we conducted a detailed analysis of feature extraction and selection. To ensure the effectiveness and diversity of the features, we approached feature extraction and selection from multiple perspectives. Addressing the issue that previous studies only focused on one aspect of either nucleotides or amino acids for feature mining, SORFPP integrates both aspects for information extraction. Through experimentation, we found that feature methods related to the calculation of sequence frequencies contribute to improved prediction performance. Consequently, we sought methods involving frequency calculation within both nucleotide and amino acid approaches. Furthermore, due to the short length of SEP sequences, traditional extraction methods might not fully capture the sequence information. Therefore, to enrich the frequency-related information, we incorporated the ESM method to extract deep features. In Chapter 3 of the paper, we compared 28 traditional amino acid features and 17 traditional nucleotide methods experimentally, resulting in the identification of 10 optimal methods, which were then combined and compared to determine the best performing approach. Ultimately, we used the t-SNE method to visualize the effectiveness of the newly integrated features in distinguishing SEP data. The detailed results can be found in Figure 3, Figure 4, Figure 5, Table 3, and Table 4 of the paper.

Question 4. Although the SORFPP model outperforms others in terms of prediction accuracy, the interpretability of the ensemble learning framework remains unclear. Explain how the model interprets or ranks feature importance, especially since SEPs are biologically significant.

Answer: Thanks for your suggestion. To enhance the interpretability of the SORFPP model, we included SHAP (SHapley Additive exPlanations) analysis in the revised manuscript, visually demonstrating the impact and significance of each feature on the model output. The SHAP analysis results indicated that features such as CTD10, QSOrder32, and APAAC14 made significant contributions to the prediction of SEPs and exhibited consistency across different species (human, mouse, rat). The high SHAP values of these features highlighted their critical role in the model predictions. Furthermore, we discussed the biological relevance of these important features. For instance, CTD10 reflects the compositional characteristics of sequences, which are closely related to the structure and function of SEPs. Meanwhile, QSOrder32 and APAAC14 capture the physicochemical properties and sequential features of amino acids, which have a significant impact on the biological functions of SEPs. The detailed results can be found in Figure 1. Refer to Figure 6 in the paper. Refer to page 27 of the paper.

Question 5. The study uses datasets from human, mouse, and rat species for training and evaluation. However, the paper should address how well SORFPP generalizes to other species or datasets not included in this analysis.

Answer: Thank you for your suggestion. To further validate the generalization capability of SORFPP, we assembled datasets from three plant species, including Arabidopsis thaliana (Ath), Glycine max (Gmax) and Physcomitrella patens (Ppatens), and tested SORFPP on these datasets. The experimental results demonstrated that SORFPP exhibited outstanding performance on the plant datasets. For instance, on the Ath dataset, the ACC reached 0.951, the AUC was 0.991, and the MCC was 0.902. Moreover, excellent performance was also achieved on the Gmax and Ppatens datasets. These outcomes indicate that SORFPP possesses strong generalization capabilities across multiple species. The specific results can be found in Table 1. Refer to Table S2 in the supporting information.

Question 6. Machine learning is well-known and has been used in previous sequence analysis studies i.e., PMID: 37649385, PMID: 36174933. Therefore, the authors are suggested to refer to more works in this description to attract a broader readership.

Answer: Thanks for your suggestion. We recognized the significance of machine learning in sequence analysis and acknowledged that our citation of related work in the current version may not have been comprehensive. In the revised manuscript, we carefully reviewed the literature in the field, including the studies mentioned by the reviewers (PMID: 37649385 and PMID: 36174933), and supplemented our references with additional works that pertained to the application of machine learning in sequence analysis. The citations were located in References [37] and [51], with the specific positions detailed in the figure below. Refer to page 7 and 28 of the paper.

Question 7. While the paper mentions limitations of alignment-based methods, a direct comparison with such methods on the same datasets would offer valuable insights into the advantages and disadvantages of using SORFPP.

Answer: Thanks for your suggestion. We acknowledged the value of directly comparing SORFPP with alignment-based methods on the same dataset. However, most previous prediction methods were aimed at identifying sORF sequences, but our study focused on the identification of SEP sequences, and many methods did not provide data for the relevant datasets. Therefore, in the revised manuscript, we included three plant datasets from the sORFPred method and used our proposed SORFPP to predict on these datasets, comparing the results with sORFPred. In the Ath dataset, SORFPP ACC and MCC were 0.951 and 0.902, respectively, sORFPred were 0.887 and 0.775, respectively. In the Gmax dataset, SORFPP ACC and MCC were 0.930 and 0.860, respectively, while sORFPred were 0.881 and 0.761, respectively. In the Ppatens dataset, SORFPP ACC and MCC were 0.898 and 0.795, respectively, whereas sORFPred were 0.842 and 0.686, respectively. The detailed results were presented in Table 10 and Figure 10 (Section 3.4). Refer to page 38 of the paper.

Question 8. The manuscript does not discuss the computational requirements or efficiency of the SORFPP model, which is crucial for high-throughput applications.

Answer: Thanks for your suggestion. SORFPP utilized the NVIDIA GeForce RTX 3070 graphics card. Regarding the computational requirements and efficiency of SORFPP, we have supplemented the processing times for three datasets in the revised manuscript. Detailed results can be found in Table 4. Refer to Table S3 in the supporting information.

Question 9. While the manuscript uses standard metrics such as accuracy, F1 score, and AUC, discussing the implications of these results in real-world scenarios, such as predicting SEPs in large-scale genomic projects, would be beneficial.

Answer: Thanks for your suggestion. We recognized that while standard metrics such as accuracy, F1 score, and AUC effectively measure the predictive performance of the SORFPP model, the significance and value of these metrics in real-world applications needed to be further elaborated. The following details the specific role and impact of these metrics in predicting SEPs in large-scale genomic projects: (1) Accuracy (ACC), in the context of large-scale genomic projects, the task of predicting SEPs typically involved millions of sequences. A model with low accuracy would result in a substantial number of incorrect predictions (false positives or false negatives), thereby increasing the cost of subsequent experimental validation. In our experiments, SORFPP achieved a high level of accuracy, indicating that it can significantly reduce the number of incorrect predictions. (2) The F1 score addresses the issue of imbalance between positive and negative samples in SEPs data, where functional SEPs were often underrepresented in number. Accuracy was utilized to evaluate model performance might obscure the model ability to predict these important, yet less abundant, positive samples. The F1 score of SORFPP indicated a better balance between precision and recall. For instance, in detecting rare but functionally significant peptides (such as SEPs encoding signaling molecules), a high F1 score ensured that these rare peptides were accurately identified without being overlooked due to data imbalance. (3) The AUC (Area Under the ROC Curve) was a critical metric in large-scale screening tasks. In genomic analysis, the classification threshold needed to be adjusted for different projects to optimize results, such as increasing the recall to screen for all possible SEPs, or increasing precision to reduce the experimental workload. The AUC value of SORFPP surpassed that of other predictors, indicating its robust classification capability, whether it was expanding coverage at low thresholds or controlling the false positive rate at high thresholds. For instance, in an actual high-throughput genomic screening project, a model with a higher AUC can ensure the optimal balance point for classification, thereby significantly enhancing prediction efficiency.

Question 10. The manuscript would be strengthened by explicitly discussing the limitations of SORFPP and suggesting future research directions.

Answer: Thanks for your suggestion. In Chapter 5 (Limitations and Future Works) of the thesis, we discussed in detail the limitations and future research directions of SORFPP, including the following points:

Dataset aspect: Although the TransLnc dataset provided a rich resource of sequences, we also recognized that with the development of next-generation sequencing technologies and the construction of more comprehensive databases, it was necessary to continue supplementing and expanding the data samples.

Feature encoding aspect: Traditional encoding methods for peptide sequences were relatively mature, but the feature encoding methods for SEPs awaited further research. We proposed a plan to integrate protein language models, such as Ankh and XProtein GLM, to delve deeper into the sequence information of SEPs.

Model optimization aspect: The process of model selection and optimization was complex, time-consuming, and resource-intensive. We discussed the future integration of automated machine learning (AutoML) tools to improve the efficiency of model training and optimization.

Question 11. Some figures and tables could benefit from more detailed captions and explanations.

Answer: Thanks for your suggestion. We recognized the vital importance of figure captions and explanations in comprehending the research content. In the revised manuscript, we have conducted a comprehensive review of all figure captions and explanations, and have added relevant details where needed to ensure that readers can easily grasp the content displayed in each figure and its scientific implications.

To Reviewer: 2

Question 1. Can you give more details about TransLnc dataset?

Answers: Thanks for your suggestion. The important role of functional peptides encoded by lncRNAs in basic cellular processes and diseases had attracted the attention of many researchers. The presence of functional peptides encoded by lncRNAs emphasizes the need to distinguish between the RNA and peptide-coding functions of lncRNAs. The basic principles of lncRNA translation had been studied. For example, many short sequences, such as N6-methyladenosine modification RNA sites (m6A), had been reported to drive the internal ribosomal entry site (IRES)-like elements that facilitate the translation of non-coding RNAs. The rapid development of whole-genome translation analysis, ribosome profiling, and mass spectrometry analysis had provided the basic conditions for identifying potential lncRNA-encoded peptides. TransLnc employed six biological methods to identify SEP sequences, including m6A, IRES, whole-genome translation analysis, ribosome profiling, and mass spectrometry analysis, thereby constructing a new comprehensive resource database. At that time, TransLnc documents approximately 583,840 peptides encoded by 33,094 lncRNAs. It integrated six types of direct and indirect evidence that could suggest the coding potential of lncRNAs, with 65.28% of peptides having at least one type of evidence. Considering the strong tissue-specific expression of lncRNAs, TransLnc allowed users to query lncRNA peptides in any of the 34 relevant tissues. Moreover,

---

## [Decision Letter · Decision Letter 1]

11 Feb 2025

PONE-D-24-29982R1SORFPP: Enhancing Rich Sequence-driven Information to Identify SEPs Based on Fused Framework on Validation DatasetsPLOS ONE

Dear Dr. Nie,

Thank you for submitting your manuscript to PLOS ONE. After careful consideration, we feel that it has merit but does not fully meet PLOS ONE’s publication criteria as it currently stands. Therefore, we invite you to submit a revised version of the manuscript that addresses the points raised during the review process. Kindly address the concern raised by Reviewer #2

We look forward to receiving your revised manuscript.

Kind regards,

Prabina Kumar Meher, Ph.D.

Academic Editor

PLOS ONE

Journal Requirements:

Reviewers' comments:

Reviewer's Responses to Questions

**Comments to the Author**

1. If the authors have adequately addressed your comments raised in a previous round of review and you feel that this manuscript is now acceptable for publication, you may indicate that here to bypass the “Comments to the Author” section, enter your conflict of interest statement in the “Confidential to Editor” section, and submit your "Accept" recommendation.

Reviewer #1: All comments have been addressed

Reviewer #2: (No Response)

2. Is the manuscript technically sound, and do the data support the conclusions?

Reviewer #1: Yes

Reviewer #2: No

3. Has the statistical analysis been performed appropriately and rigorously? 

Reviewer #1: Yes

Reviewer #2: No

4. Have the authors made all data underlying the findings in their manuscript fully available?

Reviewer #1: No

Reviewer #2: Yes

5. Is the manuscript presented in an intelligible fashion and written in standard English?

Reviewer #1: Yes

Reviewer #2: Yes

6. Review Comments to the Author

Reviewer #1: My previous comments have been addressed. My previous comments have been addressed. My previous comments have been addressed.

Reviewer #2: The manuscript develops a method called SORFPP, to predict SEPs by combining deep learning and machine learning together with protein representations. Developing web-based tool is very nice. The manuscript is written quite well, but there are still some weaknesses that need to be handled:

1- Why do authors use Categorical Boost instead of Ada Boost for instance?

2- Figure 3 colors are not clear. Can you may be make it with fewer colors?

7. PLOS authors have the option to publish the peer review history of their article (what does this mean? ). If published, this will include your full peer review and any attached files.

**Do you want your identity to be public for this peer review?** For information about this choice, including consent withdrawal, please see our Privacy Policy .

Reviewer #1: No

Reviewer #2: **Yes: ** Emre Sefer

---

## [Author Response · Author response to Decision Letter 2]

13 Feb 2025

To Reviewer: 1

Thank you for evaluating the revised manuscript. We sincerely appreciate Reviewer #1's acknowledgment that our responses have adequately addressed the previous comments, and are gratified to learn that the revisions meet the expectations. The expert feedback has been instrumental in refining this study. Substantial improvements have been implemented through additional experiments, which significantly enhanced the quality of the paper.

Should there be any remaining aspects requiring clarification, we remain fully committed to further refinements. Once again, we extend our deepest gratitude to the editorial team and reviewers for dedicating your valuable time and expertise to improve this work.

To Reviewer: 2

Question 1. Why do authors use Categorical Boost instead of Ada Boost for instance?

Answers: Thank you for your insightful question regarding our choice of Categorical Boost over AdaBoost. The selection of the CatBoost model was based on three considerations: the scale of the dataset, the feature methods, and experimental comparisons.

Firstly, the dataset used in this experiment is large in size. CatBoost is specifically designed to handle categorical features without extensive preprocessing, making it particularly suitable for datasets with a large number of categorical variables. In contrast, AdaBoost typically relies on decision trees, which do not locally optimize the encoding of categorical features, potentially leading to suboptimal splits and increased training complexity. Moreover, the feature calculation methods used in this experiment may produce sparse values. CatBoost incorporates regularization techniques, which help to mitigate overfitting. Finally, through experimental comparisons, CatBoost has shown superior performance across various evaluation metrics compared to AdaBoost.

Feature Model ACC PRE AUC MCC F1 SN SP

AN Feature (Human) CatBoost 0.876 0.875 0.957 0.753 0.878 0.881 0.872

AdaBoost 0.866 0.871 0.945 0.733 0.868 0.870 0.873

AN Feature (Mouse) CatBoost 0.867 0.859 0.948 0.734 0.864 0.869 0.865

AdaBoost 0.850 0.841 0.941 0.700 0.849 0.856 0.851

AN Feature (Rat) CatBoost 0.822 0.802 0.913 0.645 0.818 0.836 0.810

AdaBoost 0.799 0.765 0.880 0.598 0.790 0.814 0.788

Question 2. Figure 3 colors are not clear. Can you may be make it with fewer colors?

Answers: Thank you for your valuable feedback regarding Figure 3. We appreciate your suggestion to improve the clarity of colors. To address this concern, we have reduced the number of colors used in the figure while maintaining the distinction among different elements. This adjustment enhances readability and ensures better visual contrast.

Please do not hesitate to contact us if there are any question. Thanks again to the reviewers and editors for your hard work! Best wishes to you!

---

## [Editor Report · Decision Letter 2]

18 Feb 2025

SORFPP: Enhancing Rich Sequence-driven Information to Identify SEPs Based on Fused Framework on Validation Datasets

PONE-D-24-29982R2

Dear Dr. Nie,

We’re pleased to inform you that your manuscript has been judged scientifically suitable for publication and will be formally accepted for publication once it meets all outstanding technical requirements.

Kind regards,

Prabina Kumar Meher, Ph.D.

Academic Editor

PLOS ONE

---

## [Editor Report · Acceptance letter]

PONE-D-24-29982R2

PLOS ONE

Dear Dr. Nie,

I'm pleased to inform you that your manuscript has been deemed suitable for publication in PLOS ONE. Congratulations! Your manuscript is now being handed over to our production team.

Kind regards,

on behalf of

Dr. Prabina Kumar Meher

Academic Editor

PLOS ONE